# Mouse visual cortex as a limited resource system that self-learns an ecologically-general representation

**Aran Nayebi** [1,2,3]ʘ*, **Nathan C. L. Kong** [1,4,5]ʘ*, **Chengxu Zhuang** [1,3,4], **Justin L. Gardner** [1,4], **Anthony M. Norcia** [1,4], **Daniel L. K. Yamins** [1,4,6]

**1** Wu Tsai Neurosciences Institute, Stanford University, Stanford, California, United States of America, **2** Neurosciences Ph.D. Program, Stanford University, Stanford, California, United States of America, **3** McGovern Institute for Brain Research, Massachusetts Institute of Technology (MIT), Cambridge, Massachusetts, United States of America, **4** Department of Psychology, Stanford University, Stanford, California, United States of America, **5** Department of Brain and Cognitive Sciences, Massachusetts Institute of Technology (MIT), Cambridge, Massachusetts, United States of America, **6** Department of Computer Science, Stanford University, Stanford, California, United States of America

ʘ These authors contributed equally to this work.
\* anayebi@mit.edu (AN); nclkong@mit.edu (NCLK)

**Data Availability Statement:** All relevant neural data is accessible at: https://github.com/AllenInstitute/AllenSDK All source code is available at: https://github.com/neuroailab/mouse-vision.

## Abstract

Studies of the mouse visual system have revealed a variety of visual brain areas that are thought to support a multitude of behavioral capacities, ranging from stimulus-reward associations, to goal-directed navigation, and object-centric discriminations. However, an overall understanding of the mouse's visual cortex, and how it supports a range of behaviors, remains unknown. Here, we take a computational approach to help address these questions, providing a high-fidelity quantitative model of mouse visual cortex and identifying key structural and functional principles underlying that model's success. Structurally, we find that a comparatively shallow network structure with a low-resolution input is optimal for modeling mouse visual cortex. Our main finding is functional—that models trained with task-agnostic, self-supervised objective functions based on the concept of contrastive embeddings are much better matches to mouse cortex, than models trained on supervised objectives or alternative self-supervised methods. This result is very much unlike in primates where prior work showed that the two were roughly equivalent, naturally leading us to ask the question of why these self-supervised objectives are better matches than supervised ones in mouse. To this end, we show that the self-supervised, contrastive objective builds a general-purpose visual representation that enables the system to achieve better transfer on out-of-distribution visual scene understanding and reward-based navigation tasks. Our results suggest that mouse visual cortex is a low-resolution, shallow network that makes best use of the mouse's limited resources to create a light-weight, general-purpose visual system—in contrast to the deep, high-resolution, and more categorization-dominated visual system of primates.

**Funding:** A.N. is supported by the K. Lisa Yang Integrative Computational Neuroscience (ICoN) Center at MIT. N.C.L.K. is supported by the Stanford University Ric Weiland Graduate Fellowship. J.L.G. acknowledges support from the Wu Tsai Neurosciences Institute and Institute for Human-Centered AI. A.M.N. is supported by the Stanford Institute for Human Centered Artificial Intelligence. D.L.K.Y. is supported by the James S. McDonnell Foundation (Understanding Human Cognition Award Grant No. 220020469), the Simons Foundation (Collaboration on the Global Brain Grant No. 543061), the Sloan Foundation (Fellowship FG-2018-10963), the National Science Foundation (RI 1703161 and CAREER Award 1844724), the DARPA Machine Common Sense program, and hardware donation from the NVIDIA Corporation. The funders had no role in study design, data collection and analysis, decision to publish, or preparation of the manuscript.

**Competing interests:** The authors declare no competing interests.

## Author summary

Studies of mouse visual behavior have revealed a multitude of visual abilities, ranging from stimulus-reward associations, to goal-directed navigation, and object-centric discriminations. A principled system-wide model of mouse visual cortex would be useful both in providing an organizing theory for this wide spectrum of behaviors, and enabling practical technology for many model-driven studies of mouse neuroscience more broadly. However, the standard approach to creating quantitatively accurate models of primate visual cortex has been less successful with the mouse system. Here we identify critical computational features needed to capture mouse-specific neural stimulus-response patterns, and illustrate how these features can be interpreted as giving the highly resource-limited mouse brain a comparative advantage in self-learning a task-general visual representation.

## Introduction

The mouse has become an indispensable model organism in systems neuroscience, allowing unprecedented genetic and experimental control at the level of cell-type specificity in individual circuits [1]. Moreover, studies of mouse visual behavior have revealed a multitude of abilities, ranging from stimulus-reward associations, to goal-directed navigation, and object-centric discriminations [2]. Thus, the mouse animal model bridges fine-grained, low-level experimental control with high-level behavior. Understanding the principles underlying the structure and function of the mouse visual system and how it relates to more commonly studied visual organisms such as macaque, is therefore important.

Computational models are an effective tool for understanding how mouse visual cortex is capable of supporting such behaviors and for providing a normative account for its structure and function. They allow us to identify the key ingredients leading to the model with the best quantitative agreement with the neural data. We can also assess the functional similarities and differences between rodent and primate visual cortex, which otherwise would be hard to capture in the absence of a model from prior literature, beyond failures to be homologous in higher visual areas past V1. Furthermore, these models provide a natural starting point for understanding higher-level processing downstream of the visual system, such as in memory and its role during the navigation of rich visual environments [3–7]. Without an explicit model of a visual system, it is difficult to disentangle the contributions to neural response variance of the visual system from those of higher-level cognitive phenomena. Understanding the computations underlying higher cognition and motor control in rodents will therefore critically depend on an understanding of the upstream sensory areas that they depend on.

Deep convolutional neural networks (CNNs) are a class of models that have had success as predictive models of the human and non-human primate ventral visual stream (e.g., [8–13]). In contrast, such models have been poor predictors of neural responses in mouse visual cortex [14, 15]. Our hypothesis is that this failure can be understood via and be remedied by the goal-driven modeling approach [16]. This approach posits that normative models in neuroscience should pay careful attention to the objective functions (i.e., behavior), architectures (i.e., neural circuit), and data stream (i.e., visual input). These structural and functional ingredients should be finely tuned to the biology and the ecology of the organism under study.

In this work, we build a substantially improved model of mouse visual cortex by better aligning the objective function, architecture, and visual input with those of the mouse visual system. Firstly, from an objective function point of view, the primate ventral stream models

are trained in a supervised manner on ImageNet [17, 18], which is an image set containing over one million images belonging to one thousand, mostly human-relevant, semantic categories [19]. While such a dataset is an important technical tool for machine learning, it is highly implausible as a biological model particularly for rodents, who do not receive such category labels over development. Instead, we find that models trained using self-supervised, contrastive algorithms provide the best correspondence to mouse visual responses. Interestingly, this situation is different than in primates, where prior worked showed that the two were roughly equivalent [20]. Secondly, in terms of the architecture, these primate ventral visual stream models are too deep to be plausible models of the mouse visual system, since mouse visual cortex is known to be more parallel and much shallower than primate visual cortex [21–24]. By varying the number of linear-nonlinear layers in the models, we find that models with fewer linear-nonlinear layers can achieve neural predictivity performance that is better or on par with very deep models. Finally, mice are known to have lower visual acuity than that of primates [25, 26], suggesting that the resolution of the inputs to mouse models should be lower than that of the inputs to primate models. Indeed, we find that model fidelity can be improved by training them on lower-resolution images. Ultimately, the confluence of these ingredients, leads to a model, known as "Contrastive AlexNet" (first four layers), that best matches mouse visual cortex thus far.

We then address the question of why the Contrastive AlexNet is better at neural predictivity from an ecological point of view, especially the role of contrastive, self-supervised learning, which is novel and not expected by the known physiology and behavioral experiments in mouse. To address this question, we use Contrastive AlexNet to assess out-of-distribution generalization from its original training environment, including using the visual encoder as the front-end of a bio-mechanically realistic virtual rodent operating in an environment that supports spatially-extended reward-based navigation. We show that visual representations of this model lead to improved transfer performance over its supervised counterpart across environments, illustrating the congruence between the task-transfer performance and improved neural fidelity of the computational model.

Taken together, our best models of the mouse visual system suggest that it is a shallower, general-purpose system operating on comparatively low-resolution inputs. These identified factors therefore provide interpretable insight into the confluence of constraints that may have given rise to the system in the first place, suggesting that these factors were crucially important given the ecological niche in which the mouse is situated, and the resource limitations to which it is subject.

## Results

### Determining the animal-to-animal mapping transform

Prior models of mouse visual cortex can be improved by varying three ingredients to better match the biology and the ecology of mouse visual cortex. Before model development, however, we must determine the appropriate procedure by which to evaluate models. As in prior work on modeling primate visual cortex, we "map" model responses to biological responses and the ability of the model responses to recapitulate biological responses determines the model's neural fidelity [8, 9, 17, 20].

How should artificial neural network responses be mapped to biological neural responses? What firing patterns of mouse visual areas are common across multiple animals, and thus worthy of computational explanation? A natural approach would be to map artificial neural network features to mouse neural responses in the same manner that different animals can be mapped to each other. Specifically, we aimed to identify the best performing class of similarity

transforms needed to map the firing patterns of one animal's neural population to that of another, which we denote as the "inter-animal consistency". We took inspiration from methods that have proven useful in modeling human and non-human primate visual, auditory, and motor cortices [16, 27–29]. As with other cortical areas, this transform class likely cannot be so strict as to require fixed neuron-to-neuron mappings between cells. However, the transform class for each visual area also cannot be so loose as to allow an unconstrained nonlinear mapping, since the model already yields an image-computable nonlinear response.

We explored a variety of linear mapping transform classes (fit with different constraints) between the population responses for each mouse visual area (Fig 1A). The mouse visual responses to natural scenes were collected previously using both two-photon calcium imaging and Neuropixels by the Allen Institute [15, 22] from areas V1 (VISp), LM (VISl), AL (VISal), RL (VISrl), AM (VISam), and PM (VISpm) in mouse visual cortex (see number of units and specimens for each dataset in Table 1 and further details in the "Neural Response Datasets" section).

We focused on the natural scene stimuli, consisting of 118 images, each presented 50 times (i.e., 50 trials per image). For all methods, the corresponding mapping was trained on 50% of all the natural scene images, and evaluated on the remaining held-out set of images. We also included representational similarity analyses (RSA; [30]) as a baseline measure of population-wide similarity across animals, corresponding to no selection of individual units, unlike the other mapping transforms. For the strictest mapping transform (One-to-One), each target unit was mapped to the single most correlated unit in the source animal. Overall, the One-to-One mapping tended to yield the lowest inter-animal consistency among the maps considered. However, Ridge regression (L2-regularized) and Partial Least Squares (PLS) regression were more effective at the inter-animal mapping, yielding the most consistent fits across visual areas, with PLS regression providing the highest inter-animal consistency. We therefore used PLS regression in the evaluation of a candidate model's ability to predict neural responses. This mapping transform confers the additional benefit of enabling direct comparison to prior primate ventral stream results (which also used this mapping [8, 17]) in order to better understand ecological differences between the two visual systems across species.

Under all mapping transforms, a log-linear extrapolation analysis (S5 Fig) reveals that as the number of units increases, the inter-animal consistency approaches 1.0 more rapidly for the Neuropixels dataset, than the calcium imaging dataset, which were obtained from the

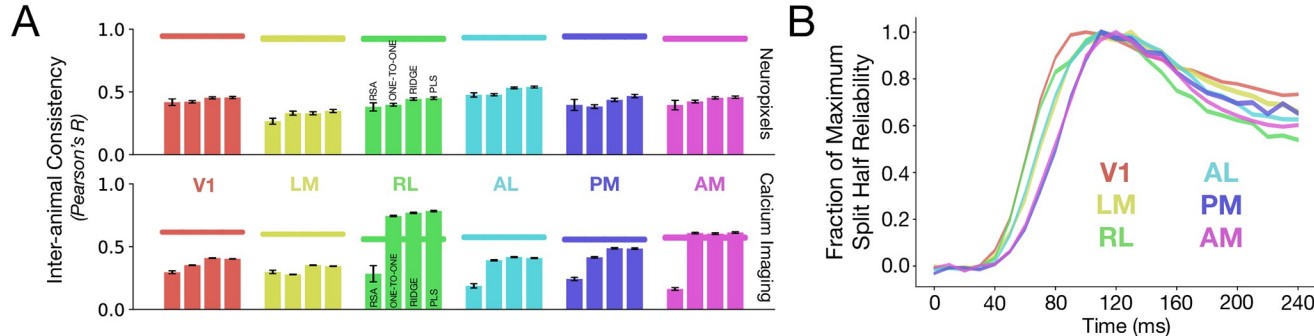

**Fig 1. Inter-animal neural response consistency across mouse visual areas. A**. Inter-animal consistency was computed using different linear maps, showing that PLS regression provides the highest consistency. Horizontal bars at the top are the median and s.e.m. of the internal consistencies of the neurons in each visual area. Refer to Table 1 for *N* units per visual area. **B**. The fraction of maximum split-half reliability is plotted as a function of time (in 10-ms time bins) for each visual area.

**Table 1. Descriptive statistics of the neural datasets.** Total number of units and specimens for each visual area for the calcium imaging and Neuropixels datasets.

| Dataset Type | Visual Area | Total Units | Total Specimens |
|---|---|---|---|
| Calcium Imaging | V1 | 7080 | 29 |
| | LM | 4393 | 24 |
| | AL | 2064 | 9 |
| | RL | 1116 | 8 |
| | AM | 847 | 9 |
| | PM | 1844 | 19 |
| Neuropixels | V1 | 442 | 8 |
| | LM | 162 | 6 |
| | AL | 396 | 6 |
| | RL | 299 | 7 |
| | AM | 257 | 7 |
| | PM | 175 | 5 |

average of the $\Delta f/F$ trace—indicating the higher reliability of the Neuropixels data across all visual areas. We further noticed a large difference between the inter-animal consistency obtained via RSA and the consistencies achieved by any of the other mapping transforms for the responses in RL of the calcium imaging dataset (green in Fig 1A). This difference, however, was not observed for responses in RL in the Neuropixels dataset. This discrepancy suggested that there was a high degree of population-level heterogeneity in the responses collected from the calcium imaging dataset, which may be attributed to the fact that the two-photon field-of-view for RL spanned the boundary between the visual and somatosensory cortex, as originally noted by de Vries et al. [15]. We therefore excluded RL in the calcium imaging dataset from further analyses, following Siegle et al. [31], who systematically compared these two datasets. Thus, this analysis provided insight into the experiments from which the data were collected, and allowed us to ascertain the level of neural response variance that is common across animals and that therefore should be "explained" by candidate models. For these reasons above, we present our main results on the newer and more reliable Neuropixels dataset, as it is in a better position to separate models—with similar results on the calcium imaging dataset presented in the Supporting Information.

## Modeling mouse visual cortex

**Building quantitatively accurate models.**   With this mapping and evaluation procedure, we can then develop models to better match mouse visual responses. The overall conclusion that the mouse visual system is most consistent with an artificial neural network model that is self-supervised, low-resolution, and comparatively shallow. This conclusion holds more generally as well on the earlier calcium imaging dataset, along with non-regression-based comparisons like RSA (cf. S2, S3 and S4 Figs). Our best models attained neural predictivity of 90% of the inter-animal consistency, much better than the prior high-resolution, deep, and task-specific model (VGG16), which attained 56.27% of this ceiling (Fig 2A; cf. rightmost column of Table 2 for the unnormalized quantities).

We also attain neural predictivity improvements over prior work [32, 33] (cf. the purple and green bars in Fig 2A), especially the latter "MouseNet" of Shi et al. [33], which attempts to map details of the mouse connectome [34, 35] onto a CNN architecture. There are two green bars in Fig 2A since we also built our own variant of MouseNet where everything is the same

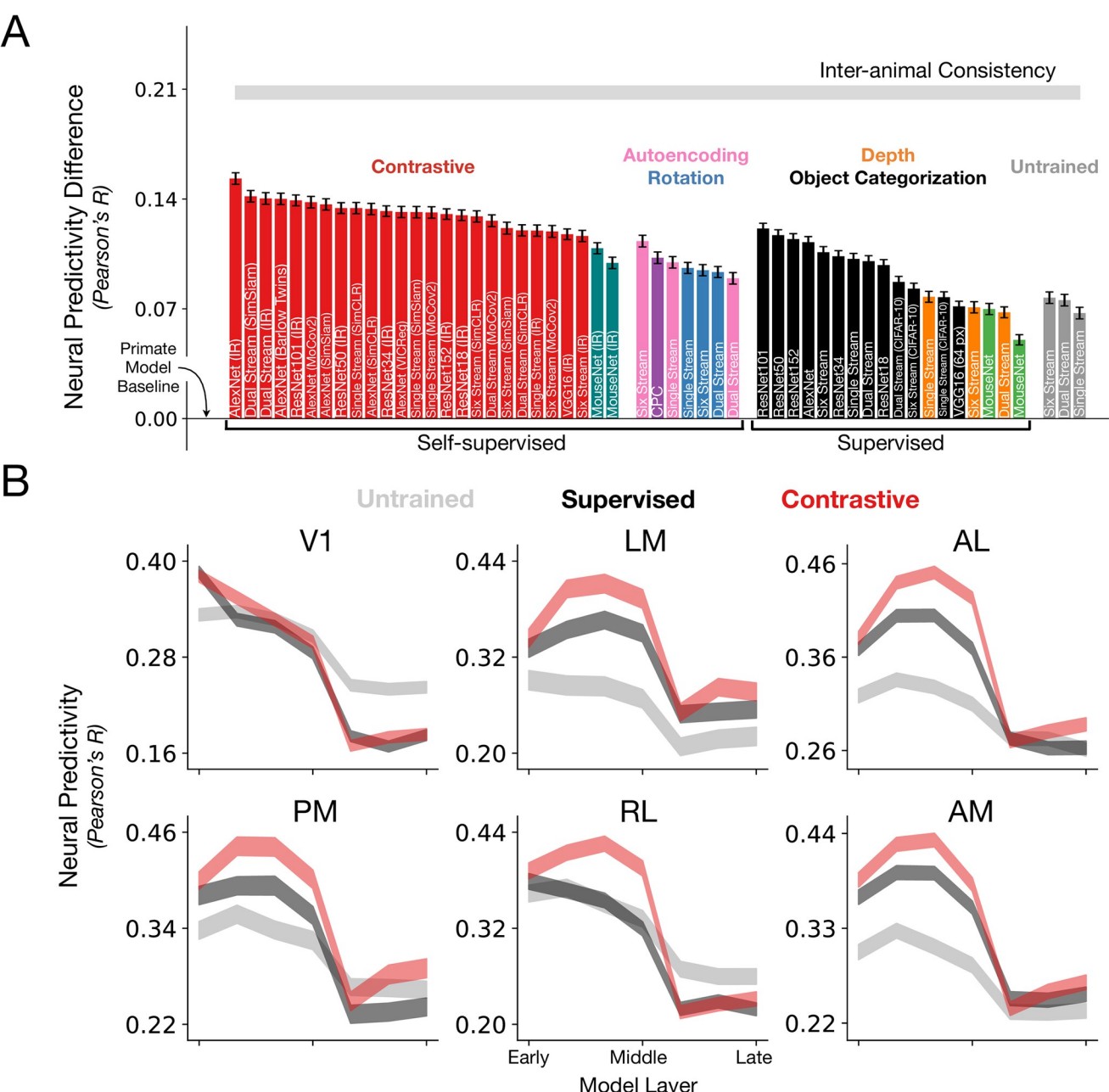

**Fig 2. Substantially improving neural response predictivity of models of mouse visual cortex. A**. The median and s.e.m. (across units) neural predictivity difference with the prior-used, primate model of Supervised VGG16 trained on 224 px inputs ("Primate Model Baseline", used in [14, 15, 36]), under PLS regression, across units in all mouse visual areas ($N$ = 1731 units in total). Absolute neural predictivity (always computed from the model layer that best predicts a given visual area) for each model can be found in Table 2. Our best model is denoted as "AlexNet (IR)" on the far left. "Single Stream", "Dual Stream", and "Six Stream" are novel architectures we developed based on the first four layers of AlexNet, but additionally incorporates dense skip connections, known from the feedforward connectivity of the mouse connectome [34, 35], as well as multiple parallel streams (schematized in S1 Fig). CPC denotes contrastive predictive coding [32, 37]. All models, except for the "Primate Model Baseline", are trained on 64 px inputs. We also note that all the models are trained using ImageNet, except for CPC (purple), Depth Prediction (orange), and the CIFAR-10-labeled black bars. **B**. Training a model on a contrastive objective improves neural predictivity across all visual areas. For each visual area, the neural predictivity values are plotted across all model layers for an untrained AlexNet, Supervised (ImageNet) AlexNet and Contrastive AlexNet (ImageNet, instance recognition)—the latter's first four layers form the best model of mouse visual cortex. Shaded regions denote mean and s.e.m. across units.

**Table 2. ImageNet top-1 validation set accuracy via linear transfer or via supervised training and neural predictivity for each model.** We summarize here the top-1 accuracy for each self-supervised and supervised model on ImageNet as well as their noise-corrected neural predictivity obtained via the PLS map (aggregated across all visual areas). These values are plotted in Fig 2C and S2 Fig. Unless otherwise stated, each model is trained and validated on 64 × 64 pixels images.

| Architecture | Objective Function | ImageNet Transfer Top-1 Accuracy | Neural Predictivity Neuropixels; Calcium Imaging |
|---|---|---|---|
| Single Stream | Untrained | 9.28% | 32.76%; 28.65% |
| | Supervised | 34.87% | 36.21%; 29.73% |
| | Autoencoder | 10.37% | 35.99%; 28.69% |
| | Depth Prediction | 18.04% | 33.79%; 27.54% |
| | RotNet | 19.72% | 35.63%; 29.27% |
| | Instance Recognition | 21.22% | 38.01%; 30.88% |
| | SimSiam | 26.48% | 39.19%; 30.48% |
| | MoCov2 | 27.63% | 39.17%; 30.30% |
| | SimCLR | 22.84% | 39.45%; 29.50% |
| Dual Stream | Untrained | 10.85% | 33.58%; 29.24% |
| | Supervised | 38.68% | 36.07%; 29.43% |
| | Autoencoder | 10.26% | 34.97%; 28.74% |
| | Depth Prediction | 19.81% | 32.81%; 27.20% |
| | RotNet | 23.29% | 35.37%; 29.15% |
| | Instance Recognition | 22.55% | 40.07%; 30.64% |
| | SimSiam | 29.21% | 40.20%; 30.60% |
| | MoCov2 | 31.00% | 38.64%; 30.33% |
| | SimCLR | 26.25% | 38.03%; 29.08% |
| Six Stream | Untrained | 11.12% | 33.74%; 29.26% |
| | Supervised | 34.15% | 36.64%; 29.79% |
| | Autoencoder | 9.27% | 37.34%; 31.12% |
| | Depth Prediction | 18.27% | 33.12%; 27.63% |
| | RotNet | 22.78% | 35.49%; 28.97% |
| | Instance Recognition | 26.49% | 37.67%; 31.18% |
| | SimSiam | 30.52% | 38.17%; 30.46% |
| | MoCov2 | 32.70% | 37.96%; 30.44% |
| | SimCLR | 28.42% | 38.92%; 29.19% |
| AlexNet | Supervised | 36.22% | 37.28%; 30.34% |
| AlexNet | Instance Recognition | 16.09% | 41.33%; 31.60% |
| AlexNet | VICReg | 32.31% | 39.20%; 30.56% |
| AlexNet | Barlow Twins | 12.38% | 40.04%; 31.42% |
| AlexNet | SimSiam | 6.84% | 39.68%; 30.95% |
| AlexNet | MoCov2 | 15.67% | 39.83%; 31.20% |
| AlexNet | SimCLR | 26.85% | 39.39%; 30.27% |
| ResNet-18 | Supervised | 53.31% | 35.82%; 28.93% |
| ResNet-18 | Instance Recognition | 30.75% | 38.99%; 30.11% |
| ResNet-34 | Supervised | 58.24% | 36.38%; 28.42% |
| ResNet-34 | Instance Recognition | 32.91% | 39.26%; 30.93% |
| ResNet-50 | Supervised | 63.57% | 37.72%; 30.70% |
| ResNet-50 | Instance Recognition | 38.49% | 39.45%; 31.12% |
| ResNet-101 | Supervised | 64.53% | 38.15%; 31.05% |
| ResNet-101 | Instance Recognition | 39.43% | 39.94%; 31.60% |
| ResNet-152 | Supervised | 65.21% | 37.74%; 30.67% |
| ResNet-152 | Instance Recognition | 40.08% | 39.07%; 31.30% |
| VGG16 | Supervised | 58.32% | 33.19%; 27.09% |

(*Continued*)

**Table 2.** (Continued)

| Architecture | Objective Function | ImageNet Transfer Top-1 Accuracy | Neural Predictivity Neuropixels; Calcium Imaging |
|---|---|---|---|
| VGG16 (224 px) | Supervised | 71.59% | 26.03%; 20.40% |
| VGG16 | Instance Recognition | 25.33% | 37.78%; 28.66% |
| MouseNet of [33] | Supervised | 37.14% | 31.05%; 25.89% |
| MouseNet of [33] | Instance Recognition | 21.21% | 35.96%; 30.06% |
| MouseNet Variant | Supervised | 39.37% | 33.02%; 26.53% |
| MouseNet Variant | Instance Recognition | 26.85% | 36.89%; 28.80% |
| CPC of [32] | Contrastive Predictive Coding | N/A | 36.28%; 29.67% |

except that image categories are read off of at the penultimate layer of the model (rather than the concatenation of the earlier layers as originally proposed). We thought this might aid the original MouseNet's task performance and neural predictivity, since it can be difficult to train linear layers when the input dimensionality is very large. Our best models also outperform the neural predictivity of "MouseNet" even when it is trained with a self-supervised objective (left-most red vs. "MouseNet" teal bars in Fig 2A). This is another conceptual motivation for our structural-and-functional goal-driven approach, as the higher-level constraints are easier to interrogate than to incorporate and assume individual biological details, as this can be a very under constrained procedure.

In the subsequent subsections, we distill the three factors that contributed to models with improved correspondence to the mouse visual areas: objective function, input resolution, and architecture.

**Objective function: Training models on self-supervised, contrastive objectives, instead of supervised objectives, improves correspondence to mouse visual areas.** The success in modeling the human and the non-human primate visual system has largely been driven by convolutional neural networks trained in a supervised manner on ImageNet [19] to perform object categorization [9, 13]. This suggests that models trained with category-label supervision learn useful visual representations that are well-matched to those of the primate ventral visual stream [17, 20]. Thus, although biologically implausible for the rodent, category-label supervision is a useful starting point for building baseline models and indeed, models trained in this manner are much improved over the prior primate model of VGG16 (black bars in Fig 2A). We refer to this model as the "Primate Model Baseline", since although many different models have been used to predict neural activity in the primate ventral stream, this model in particular was the *de facto* CNN used in the initial goal-driven modeling studies of mouse visual cortex [14, 15, 36]. This choice also helps explicitly illustrate how the three factors that we study, which deviate in visual acuity, model depth, and functional objective from the primate ventral stream, quantitatively improve these models greatly.

These improved, category-label-supervised models, however, cannot be an explanation for how the mouse visual system developed in the first place. In particular, it is unclear whether or not rodents can perform well on large-scale object recognition tasks when trained, such as tasks where there are hundreds of labels. For example, they obtain approximately 70% on a two-alternative forced-choice object classification task [38]. Furthermore, the categories of the ImageNet dataset are human-centric and therefore not relevant for rodents In fact, the affordances of primates involve being able to manipulate objects flexibly with their hands (unlike rodents). Therefore, the ImageNet categories may be more relevant to non-human primates as an ethological proxy than it is to rodents.

We therefore turned to more ecological supervision signals and self-supervised objective functions, where high-level category labeling is not necessary. These objectives could possibly lead to models with improved biological plausibility and may provide more general goals for the models, based on natural image statistics, beyond the semantic specifics of (human-centric) object categorization. Among the more ecological *supervised* signals, we consider categorization of comparatively lower-variation and lower-resolution images with fewer labels (CIFAR-10-labeled black bars in Fig 2A; [39]), and depth prediction (orange bars in Fig 2A; [40]), as a visual proxy for whisking [41]. Here, we note that in Fig 2A, all models except for those indicated by CIFAR-10 (black), CPC (purple), or Depth Prediction (orange), are trained on ImageNet images. Therefore, even if we remove the category labels from training, many of the images *themselves* (mainly from both ImageNet and CIFAR-10) are human-centric so future models could be trained using images that are more plausible for mice.

Turning to *self-supervision*, early self-supervised objectives include sparse autoencoding (pink bars in Fig 2A; [42]), instantiated as a sparsity penalty in the latent space of the image reconstruction loss, which has been shown to be successful at producing Gabor-like functions reminiscent of the experimental findings in the work of Hubel and Wiesel [43]. These objectives, however, have not led to quantitatively accurate models of higher visual cortex [8, 20]. Further developments in computer vision have led to other self-supervised algorithms, motivated by the idea that "non-semantic" features are highly related to the higher-level, semantic features (i.e., category labels), such as predicting the rotation angle of an image (blue bars in Fig 2A; [44]). Although these objectives are very simple, models optimized on them do not result in "powerful" visual representations for downstream tasks. We trained models on these objectives and showed that although they improve neural predictivity over the prior primate model (VGG16), they do not outperform category-label-supervised models (Fig 2A; compare pink, blue, and orange with black).

Further developments in self-supervised learning provided a new class of *contrastive* objectives. These objectives are much more powerful than prior self-supervised objectives described above, as it has been shown that models trained with contrastive objectives leads to visual representations that can support strong performance on downstream object categorization tasks. At a high level, the goal of these contrastive objectives is to learn a representational space where embeddings of augmentations for one image (i.e., embeddings for two transformations of the *same* image) are more "similar" to each other than to embeddings of other images. We trained models using the family of contrastive objective functions including: Instance Recognition (IR; [45]), a Simple Framework for Contrastive Learning (SimCLR; [46]), Momentum Contrast (MoCov2; [47]), Simple Siamese representation learning (SimSiam; [48]), Barlow Twins [49], and Variance-Invariance-Covariance Regularization (VICReg; [50]). Note that we are using the term "contrastive" broadly to encompass methods that learn embeddings which are robust to augmentations, even if they do not explicitly rely on negative batch *examples*—as they have to contrast against something to avoid representational collapse. For example, SimSiam relies on asymmetric representations via a stop gradient; Barlow Twins relies on regularizing with the cross correlation matrix's off diagonal elements; and VICReg uses the variance and covariance of each embedding to ensure samples in the batch are different. Models trained with these contrastive objectives (red bars in Fig 2A) resulted in higher neural predictivity across all the visual areas than models trained on supervised object categorization, depth prediction, and less-powerful self-supervised algorithms (black, orange, purple, pink, and blue bars in Fig 2A).

We hone in on the contribution of the contrastive objective function (over a supervised objective, and with a fixed dataset of ImageNet) to neural predictivity by fixing the architecture to be AlexNet, while varying the objective function (shown in S9 Fig left). We find that across

all objective functions, training AlexNet using instance recognition leads to the highest neural predictivity. We additionally note that the improvement in neural predictivity extends *beyond* the augmentation used for each objective function, as shown in S7 Fig. When the image augmentations intended for contrastive losses is used with supervised losses, neural predictivity does not improve. Across all the visual areas, there is an improvement in neural predictivity simply by using a powerful contrastive algorithm (red vs. black in Fig 2B and S9 Fig left). Not only is there an improvement in neural predictivity, but also an improvement in hierarchical correspondence to the mouse visual hierarchy. Using the brain hierarchy score developed by Nonaka et al. [51], we observe that contrastive models outperform their supervised counterparts in matching the mouse visual hierarchy (S10 Fig). The first four layers of Contrastive AlexNet (red; Fig 2B), where neural predictivity is maximal across visual areas, forms our best model for mouse visual cortex.

**Data stream: Training models on images of lower resolution improves correspondence to mouse visual areas.**   The visual acuity of mice is known to be lower than the visual acuity of primates [25, 26]. Thus, more accurate models of mouse visual cortex must be trained and evaluated at image resolutions that are lower than those used in the training of models of the primate visual system. We investigated how neural predictivity of two strong *contrastive* models varied as a function of the image resolution at which they were trained.

Two models were used in the exploration of image resolution's effects on neural predictivity. Contrastive AlexNet was trained with image resolutions that varied from $64 \times 64$ pixels to $224 \times 224$ pixels, as $64 \times 64$ pixels was the minimum image size for AlexNet due to its architecture. The image resolution upper bound of $224 \times 224$ pixels is the image resolution that is typically used to train neural network models of the primate ventral visual stream [17]. We also investigated image resolution's effects using a novel model architecture we developed, known as "Contrastive StreamNet", as its architecture enables us to explore a lower range of image resolutions than the original AlexNet. This model was based on the first four layers of AlexNet, but additionally incorporates dense skip connections, known from the feedforward connectivity of the mouse connectome [34, 35], as well as multiple parallel streams (schematized in S1 Fig). We trained it using a contrastive objective function (instance recognition) at image resolutions that varied from $32 \times 32$ pixels to $224 \times 224$ pixels.

Training models using resolutions lower than what is used for models of primate visual cortex improves neural predictivity across all visual areas, but not beyond a certain resolution, where neural predictivity decreases (Fig 3). Although the input resolution of $64 \times 64$ pixels may not be optimal for every architecture, it was the resolution that we used to train all the models. This was motivated by the observation that the upper bound on mouse visual acuity is 0.5 cycles / degree [25], corresponding to 2 pixels / cycle × 0.5 cycles / degree = 1 pixel / degree. Prior retinotopic map studies [52] estimate a visual coverage range in V1 of 60–90 degrees, and we found $64 \times 64$ pixels to be roughly optimal for the models (Fig 3) and was also used by Shi et al. [36], by Bakhtiari et al. [32], and in the MouseNet of Shi et al. [33]. Although downsampling training images is a reasonable proxy for the mouse retina, as has been done in prior modeling work, more investigation into appropriate image transformations may be needed.

Overall, these data show that optimization using a simple change in the image statistics (i.e., data stream) is crucial to obtain improved models of mouse visual encoding. This suggests that mouse visual encoding is the result of "task-optimization" at a lower resolution than what is typically used for primate ventral stream models.

**Architecture: Shallow models suffice for improving correspondence to mouse visual cortex.**   Anatomically, the mouse visual system has a shallower hierarchy relative to that of the primate visual system (see, e.g., [21–24]). Furthermore, the reliability of the neural responses for each visual area provides additional support for the relatively shallow functional

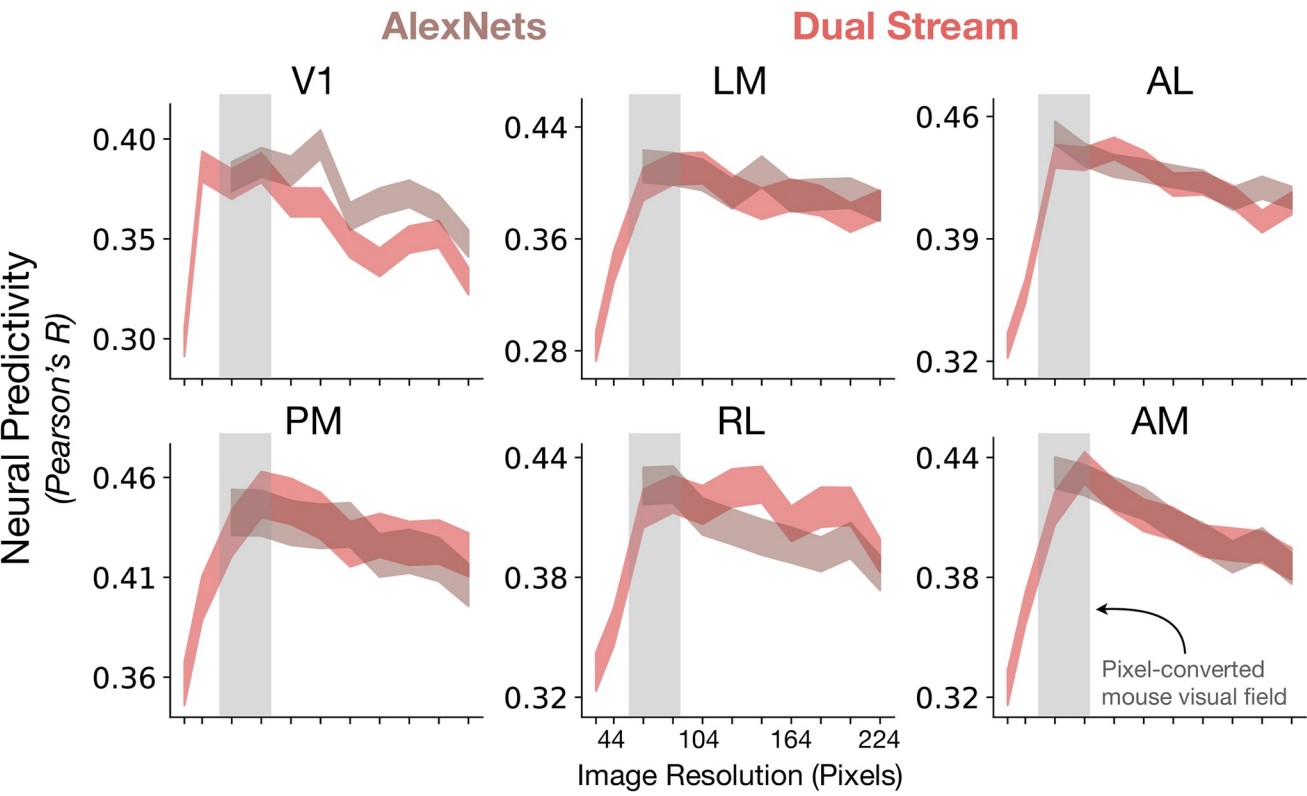

**Fig 3. Lower-resolution training leads to improved neural response predictivity (Neuropixels dataset).** Contrastive AlexNet and Contrastive StreamNet (Dual Stream) were trained using images of increasing resolutions. For AlexNet, the minimum resolution was 64 × 64 pixels. For Dual StreamNet, the minimum resolution was 32 × 32 pixels. The median and s.e.m. neural predictivity across all units of all visual areas is plotted against the image resolution at which the models were trained. Reducing the image resolution (but not beyond a certain point) during representation learning improves the match to all the visual areas. The conversion from image resolution in pixels to visual field coverage in degrees is under the assumption that the upper bound on mouse visual acuity is 0.5 cycles per degree [25] and that the Nyquist limit is 2 pixels per cycle, leading to a conversion ratio of 1 pixel per degree.

hierarchy (Fig 1B). Thus, more biologically-plausible models of mouse visual cortex should have fewer linear-nonlinear layers than those of primate visual cortex. By plotting a model's neural predictivity against its number of linear-nonlinear operations, we indeed found that very deep models do not outperform shallow models (i.e., models with less than 12 linear-non-linear operations), despite changes across loss functions and input resolutions (Fig 4). In addition, if we fix the objective function to be "instance recognition", we can clearly observe that AlexNet, which consists of eight linear-nonlinear layers, has the highest neural predictivity compared to ResNets and VGG16 (S9 Fig right). Furthermore, models with only four convolutional layers (StreamNets; Single, Dual, or Six Streams) perform as well as or better than models with many more convolutional layers (e.g., compare Dual Stream with ResNet101, ResNet152, VGG16, or MouseNets in S9 Fig right). This observation is in direct contrast with prior observations in the primate visual system whereby networks with fewer than 18 linear-nonlinear operations predict macaque visual responses less well than models with at least 50 linear-nonlinear operations [17].

## Mouse visual cortex as a general-purpose visual system

Our results show that a model optimized on a self-supervised contrastive objective the most quantitatively accurate model of the mouse visual system. However, unlike supervised object

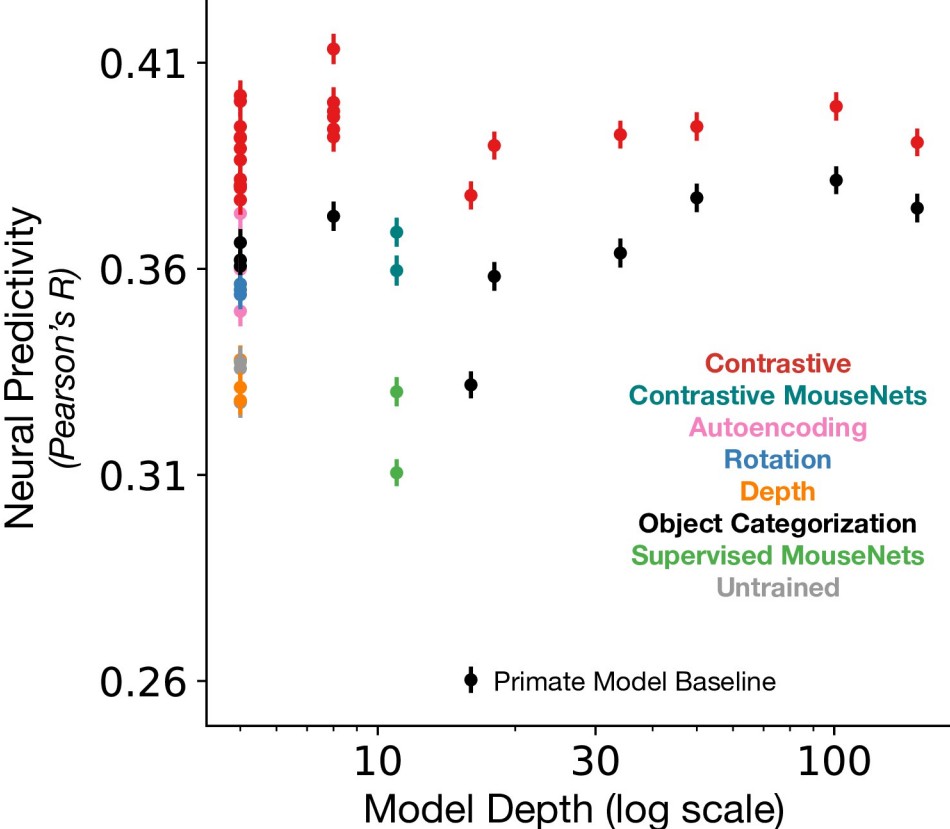

**Fig 4. Neural predictivity vs. model depth.** A model's median neural predictivity across all units from each visual area is plotted against its depth (number of linear-nonlinear layers; in log-scale). Models with fewer linear-nonlinear layers can achieve neural predictivity performances that outperform or are on par with those of models with many more linear-nonlinear layers. The "Primate Model Baseline" denotes a supervised VGG16 trained on 224 px inputs, used in prior work [14, 15, 36].

categorization or other more-classical forms of self-supervised learning such as sparse autoencoding, whose ecological function is directly encoded in the loss function (e.g., predator recognition or metabolically-efficient dimension reduction), the functional utility of self-supervision via a contrastive objective is not as apparent. This naturally raises the question of what behavioral function(s) optimizing a contrastive self-supervision objective might enable for the mouse from an ecological fitness viewpoint.

Analyzing the spectrum of models described in the above section, we first observed that performance on ImageNet categorization is uncorrelated with improved neural predictivity for mouse visual cortex (Fig 5; cf. middle column of Table 2 for the exact ImageNet performance values), unlike the well-known correlation in primates (Fig 5; inset). In seeking an interpretation of the biological function of the contrastive self-supervised objective, we were thus prompted to consider behaviors beyond object-centric categorization tasks. We hypothesized that since self-supervised loss functions are typically most effective in task-agnostic stimulus domains where goal-specific labels are unavailable, optimizing for such an objective might enable the rodent to *transfer* well to complex multi-faceted ecological tasks in novel domains affording few targeted opportunities for hyper-specialization.

To test this hypothesis, we used a recently-developed "virtual rodent" framework (adapted from Merel et al. [53] and also used by Lindsay et al. [54]), in which a biomechanically-

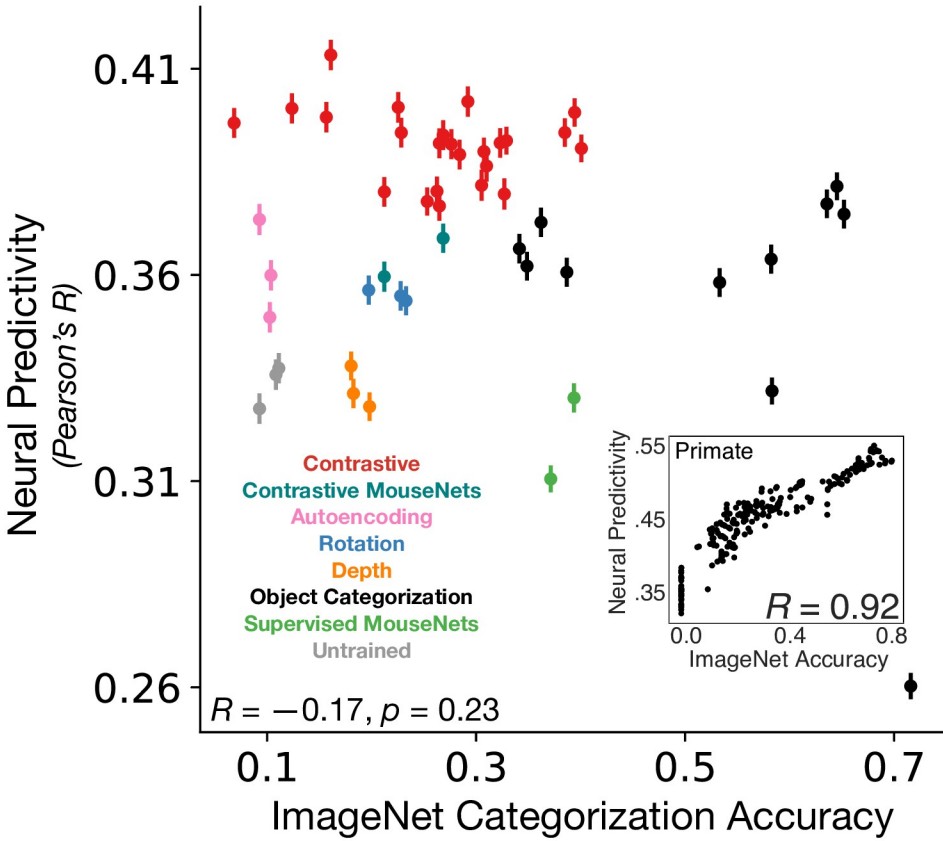

**Fig 5. Neural predictivity is *not* correlated with object categorization performance on ImageNet.** Each model's (transfer or supervised) object categorization performance on ImageNet is plotted against its median neural predictivity across all units from all visual areas. All ImageNet performance values can be found in Table 2. **Inset**. Primate ventral visual stream neural predictivity from BrainScore is correlated with ImageNet categorization accuracy (adapted from Schrimpf et al. [17]). This relationship is in stark contrast to our finding in mouse visual cortex where higher ImageNet categorization accuracy is *not* associated with higher neural predictivity. Color scheme as in Fig 2A. See S2 Fig. for neural predictivity on the calcium imaging dataset.

validated mouse model is placed in a simulated 3D maze-like environment (Fig 6A). The primary purpose of the experiments with the virtual rodent are not to necessarily make a specific statement about rodent movement repertoires, but mainly to how well our self-supervised visual encoder enables control of a high-dimensional body with high-dimensional continuous inputs—a problem that many (if not all) animals have to solve. Of course, given that we were trying to better understand why self-supervised methods are better predictors of specifically mouse visual cortical neurons, we wanted a reasonable ecological task for that species (e.g., navigation) and that its affordances were somewhat similar to that of an actual rodent via the biomechanical realism of its body. In particular, if we used our shallow, lower acuity, self-supervised visual encoder for controlling a virtual monkey simulation for a task that a monkey is adapted to (e.g., object manipulation), we would not expect this to work as well, given that such a task likely requires high visual acuity and good object recognition abilities at a minimum.

We used the model that best corresponds to mouse visual responses as the visual system of the simulated mouse, coupled to a simple actor-critic reinforcement-learning architecture (Fig 6B). We then trained the simulated mouse in several visual contexts with varying objective

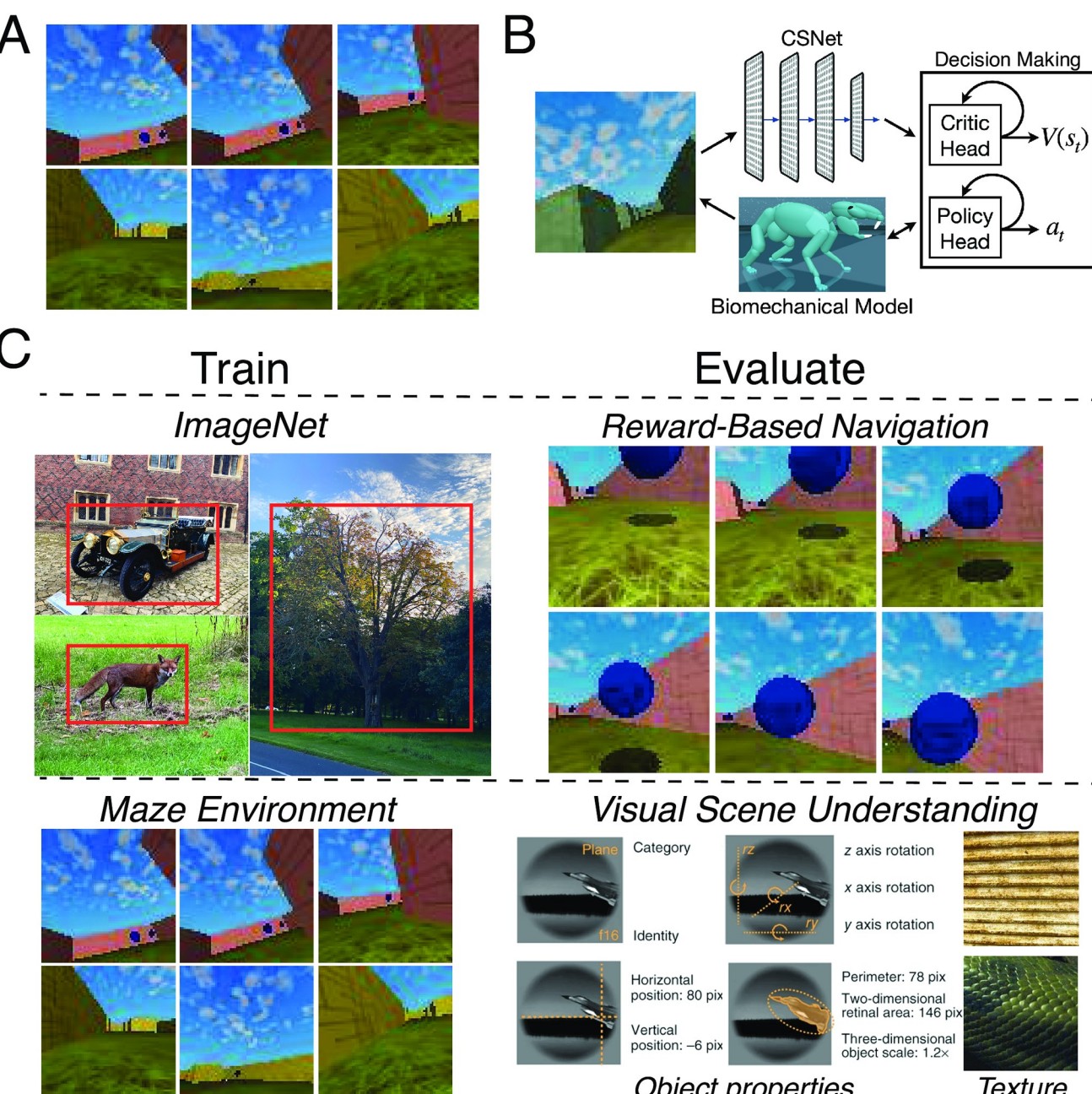

**Fig 6. Evaluating the generality of learned visual representations. A**. Each row shows an example of an episode used to train the reinforcement learning (RL) policy in an offline fashion [56]. The episodes used to train the virtual rodent (adapted from Merel et al. [53]) were previously generated and are part of a larger suite of RL tasks for benchmarking algorithms [57]. In this task ("DM Locomotion Rodent"), the goal of the agent is to navigate the maze to collect as many rewards as possible (blue orbs shown in the first row). **B**. A schematic of the RL agent. The egocentric visual input is fed into the visual backbone of the model, which is fixed to be the first four convolutional layers of either the contrastive or the supervised variants of AlexNet. The output of the visual encoder is then concatenated with the virtual rodent's proprioceptive inputs before being fed into the (recurrent) critic and policy heads. The parameters of the visual encoder are *not* trained, while the parameters of the critic head and the policy head are trained. Virtual rodent schematic from Fig 1B of Merel et al. [53]. **C**. A schematic of the out-of-distribution generalization procedure. Visual encoders are either trained in a supervised or self-supervised manner on ImageNet [19] or the Maze environment [57], and then evaluated on reward-based navigation or datasets consisting of object properties (category, pose, position, and size) from Hong et al. [55], and different textures [58].

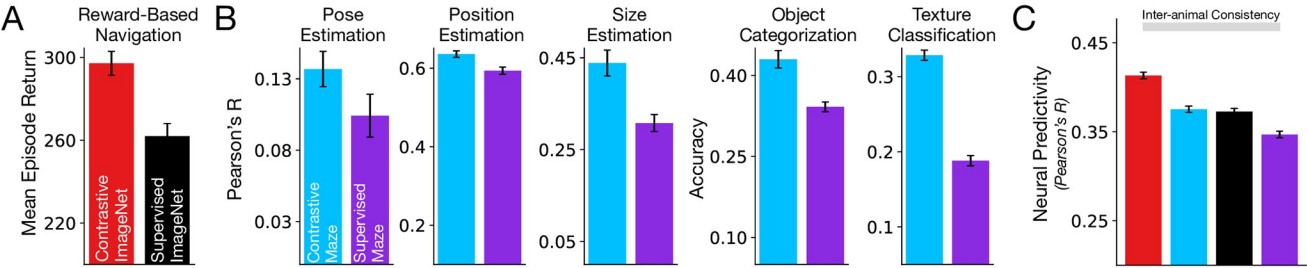

**Fig 7. Self-supervised, contrastive visual representations better support transfer performance on downstream, out-of-distribution tasks.** Models trained in a contrastive manner (using either ImageNet or the egocentric maze inputs; red and blue respectively) lead to better transfer on out-of-distribution downstream tasks than models trained in a supervised manner (i.e., models supervised on labels or on rewards; black and purple respectively). **A**. Models trained on ImageNet, tested on reward-based navigation. **B**. Models trained on egocentric maze inputs ("Contrastive Maze", blue) or supervised on rewards (i.e., reward-based navigation; "Supervised Maze", purple), tested on visual scene understanding tasks: pose, position, and size estimation, and object and texture classification. **C**. Median and s.e.m. neural predictivity across units in the Neuropixels dataset.

functions, and evaluated those models' ability to *transfer* to a variety of tasks in novel environments, including both a reward-based navigation task, as well as several object-centric visual categorization and estimation tasks (Fig 6C).

We first evaluated a simulated mouse whose visual system was pretrained with ImageNet images in terms of its ability to transfer to reward-based navigation in the Maze environment, training just the reinforcement learning portion of the network on the navigation task with the pretrained visual system held constant. As a supervised control, we performed the same transfer training procedure using a visual front-end created using supervised (ImageNet) object categorization pretraining. We found that the simulated mouse with the contrastive self-supervised visual representation was able to reliably obtain substantially higher navigation rewards than its counterpart with category-supervised visual representation (Fig 7A; cf. Table 3 for the exact quantities of rewards attained).

Conversely, we trained the simulated mouse directly in the Maze environment. In one variant, we trained the visual system on the contrastive self-supervised objective (but on the images of the Maze environment). In a second variant, we trained the agent end-to-end on the reward navigation task itself—the equivalent of "supervision" in the Maze environment. We then tested both models on their ability to transfer to out-of-sample visual categorization and estimation tasks. Again, we found that the self-supervised variant transfers significantly better than its "supervised" counterpart (blue vs. purple bars in Fig 7B).

Returning to analyses of neural predictivity, we found that both self-supervised models (trained in either environment) were better matches to mouse visual cortex neurons than the supervised counterparts in their respective environments (red and blue vs. black and purple in Fig 7C). This result illustrates the congruence between general out-of-distribution task-transfer performance and improved neural fidelity of the computational model.

**Table 3. Average reward and s.e.m. across 600 episodes obtained by the RL agent using each of the visual backbones.**

| Visual Backbone | Reward |
|---|---|
| Contrastive ImageNet | 297 ± 6 |
| Supervised ImageNet | 262 ± 6 |
| Contrastive Maze | 412 ± 5 |
| Supervised Maze | 462 ± 6 |

Furthermore, Lindsay et al. [54] found that less powerful self-supervised representation learners such as CPC and autoencoding did not match mouse visual responses as well as their RL-trained counterpart (in terms of representational similarity). This is consistent with our finding that CPC and autoencoding in Fig 3A themselves do not match neural responses as well as contrastive self-supervised methods (red vs. pink and purple bars).

It is also noteworthy that, comparing models by training environment rather than objective function, the ImageNet-trained models produce more neurally-consistent models than the Maze-trained models, both for supervised and self-supervised objectives (red vs. blue and black vs. purple in Fig 7C). Using the contrastive self-supervised objective function is by itself enough to raise the Maze-trained model to the predictivity level of the ImageNet-trained supervised model, but the training environment makes a significant contribution. This suggests that while the task domain and biomechanical model of the Maze environment are realistic, future work will likely need to improve the realism of the simulated image distribution.

Overall, these results suggest that contrastive embedding methods have achieved a generalized improvement in the quality of the visual representations they create, enabling a diverse range of visual behaviors, providing evidence for their potential as computational models of mouse visual cortex. While we expect the calculation of rewards to be performed *outside* of the visual system, we expect the mouse visual system would support the visual aspects of the transfer tasks we consider, as we find that higher model areas best support these scene understanding transfer tasks (S8 Fig). This is not unlike the primate, where downstream visual areas support a variety of visual recognition tasks [55].

## Discussion

In this work, we showed that comparatively shallow architectures trained with contrastive objectives operating on lower-resolution images most accurately predict static-image-evoked neural responses across multiple mouse visual areas, surpassing the predictive power of supervised methods and approaching the inter-animal consistency. The fact that these goal-driven constraints lead to a better match to visual responses, even in "behaviorally-free" data where mice are passively viewing stimuli, suggests that these constraints may be good descriptions of the evolutionary and the developmental drivers of mouse visual cortical structure and function.

In the primate ventral visual stream, models trained on contrastive objectives led to neural predictivity performance that was *on par* with that of supervised models [20], suggesting they are a more ecologically-valid proxy for a categorization-specialized system. This is in stark contrast with our findings in our models of mouse visual cortex—we found that models trained on contrastive objectives *substantially surpassed* the neural predictivity of their supervised counterparts. We investigated the advantages that contrastive objectives might confer over supervised objectives for mouse visual representations and found that they provide representations that are generally improved over those obtained by supervised methods. The improvement in generality of contrastive models enabled better transfer to a diverse range of downstream behaviors in novel, *out of distribution* environments, including reward-based navigation in egocentric maze environments and visual scene understanding.

As mentioned previously, the goal-driven modeling approach allows us to understand the principles that govern the system under study and further allows for direct comparisons across systems. Our high-fidelity model of mouse visual cortex and the principles underlying its construction can be compared with models of primate visual cortex. While the primate ventral visual stream is well-modeled by a deep hierarchical system and object-category learning, mouse as a model visual system has not had such a coherent account heretofore. Our results

demonstrate that both primate and rodent visual systems are highly constrained, albeit for different functional purposes. The results summarized above further suggest that mouse visual cortex is a light-weight, shallower, low-resolution, and general-purpose visual system in contrast to the deep, high-resolution, and more categorization-dominated visual system of primates, suggested by prior work [20].

Although we have made progress in modeling the mouse visual system in three core ways (the choice of objective function, data stream, and architecture class), some limitations remain both in the modeling and in the neural data.

On the architectural front, our focus in this work was on feedforward models, but there are many feedback connections from higher visual areas to lower visual areas [21]. Incorporating these architectural motifs into our models and training these models using dynamic inputs may be useful for modeling temporal dynamics in mouse visual cortex, as has been recently done in primates [29, 59, 60].

By incorporating recurrent connections in the architectures, we can probe the functionality of these feedback connections using self-supervised loss functions in scenarios with temporally-varying, dynamic inputs. For example, given that powerful self-supervised methods obtained good visual representations of static images, it would be interesting to explore a larger spectrum of self-supervised signals operating on dynamic inputs, such as in the context of forward prediction (e.g., [61–63]).

Constraining the input data so that they are closer to those received by the mouse visual system was important for improved neural fidelity. Our resizing (i.e., downsampling) of the images to be smaller during training acted as a proxy for low-pass filtering. We believe that future work could investigate other appropriate low-pass filters and ecologically-relevant pixel-level transformations to apply to the original image or video stream [64, 65].

Our inter-animal consistency analyses make recommendations for qualities of experiments that are likely to provide data that will be helpful in more sharply differentiating models. Under all mapping functions, a log-linear extrapolation analysis revealed that as the number of units increases, the inter-animal consistency approaches one more rapidly for the Neuropixels dataset, than for the calcium imaging dataset, indicating the higher reliability of the Neuropixels data (S5 Fig). Moreover, when assessing inter-animal consistency, correlation values between animals were significantly higher for the training set than for the test set, indicating that the number of stimuli could be enlarged to close this generalization gap (S6A Fig). As a function of the number of stimuli, the test set inter-animal consistencies steadily increased, and would likely continue to increase substantially if the dataset had more stimuli (S6B Fig). Thus, while much focus in experimental methods has been on increasing the number of neurons in a dataset [66], our analyses indicate that increasing the number of stimuli may drastically improve model identification. Doing so would likely raise the inter-animal consistency, providing substantially more dynamic range for separating models in terms of their ability to match the data, potentially allowing us to obtain more specific conclusions about which circuit structure(s) [67, 68] and which (combinations of) objectives (e.g., [45, 46, 48]) best describe mouse visual cortex.

We endeavor that our work in modeling mouse visual cortex will meaningfully drive future experimental and computational studies in mice of other sensory systems and of visually-guided behaviors. The input-domain-agnostic nature of these contrastive objectives suggest the tantalizing possibility that they might be used in other sensory systems, such as barrel cortex or the olfactory system. By building high-fidelity computational models of sensory cortex, we believe that they can be integrated with models of higher-order systems (e.g., medial temporal lobe), with the goal of providing us with greater insight into how sensory experience contributes to adaptive or maladaptive behaviors.

## Methods

### Neural response datasets

We used the Allen Brain Observatory Visual Coding dataset [15, 22] collected using both two-photon calcium imaging and Neuropixels from areas V1 (VISp), LM (VISl), AL (VISal), RL (VISrl), AM (VISam), and PM (VISpm) in mouse visual cortex. We focused on the natural scene stimuli, consisting of 118 images, each presented 50 times (i.e., 50 trials per image).

We list the number of units and specimens for each dataset in Table 1, after units are selected, according to the following procedure: For the calcium imaging data, we used a similar unit selection criterion as in Conwell et al. [18], where we sub-selected units that attain a Spearman-Brown corrected split-half consistency of at least 0.3 (averaged across 100 bootstrapped trials), and whose peak responses to their preferred images are not significantly modulated by the mouse's running speed during stimulus presentation ($p > 0.05$).

For the Neuropixels dataset, we separately averaged, for each specimen and each visual area, the temporal response (at the level of 10-ms bins up to 250 ms) on the largest contiguous time interval when the median (across the population of units in that specimen) split-half consistency reached at least 0.3. This procedure helps to select the most internally-consistent units in their temporally-averaged response, and accounts for the fact that different specimens have different time courses along which their population response becomes reliable.

Finally, after subselecting units according to the above criteria for both datasets, we only keep specimens that have at least the 75th percentile number of units among all specimens for that given visual area. This final step helped to ensure we have enough internally-consistent units per specimen for the inter-animal consistency estimation (derived in the "Inter-Animal Consistency Derivation" section).

### Noise-corrected neural predictivity

**Linear regression.** When we perform neural fits, we choose a random 50% set of natural scene images (59 images in total) to train the regression, and the remaining 50% to use as a test set (59 images in total), across ten train-test splits total. For Ridge regression, we use an $\alpha = 1$, following the `sklearn.linear_model` convention. PLS regression was performed with 25 components, as in prior work (e.g., [8, 17]). When we perform regression with the One-to-One mapping, as in Fig 1B, we identify the top correlated (via Pearson correlation on the training images) unit in the source population for each target unit. Once that source unit has been identified, we then fix it for that particular train-test split, evaluated on the remaining 50% of images.

Motivated by the justification given in the "Inter-Animal Consistency Derivation" section for the noise correction in the inter-animal consistency, the noise correction of the model to neural response regression is a special case of the quantity defined in the "Multiple Animals" section, where now the source animal is replaced by model features, separately fit to each target animal (from the set of available animals $\mathcal{A}$). Let $L$ be the set of model layers, let $r^\ell$ be the set of model responses at model layer $\ell \in L$, $M$ be the mapping, and let s be the trial-averaged pseudo-population response

$$\max_{\ell \in L} \text{median} \bigoplus_{B \in \mathcal{A}} \left\langle \frac{\text{Corr}\left(M(r^\ell_{\text{train}}; s^B_{1,\text{train}})_{\text{test}}, s^B_{2,\text{test}}\right)}{\sqrt{\widetilde{\text{Corr}}(M(r^\ell_{\text{train}}; s^B_{1,\text{train}})_{\text{test}}, M(r^\ell_{\text{train}}; s^B_{2,\text{train}})_{\text{test}}) \times \widetilde{\text{Corr}}(s^B_{1,\text{test}}, s^B_{2,\text{test}})}} \right\rangle,$$

where the average is taken over 100 bootstrapped split-half trials, $\oplus$ denotes concatenation of units across animals $B \in \mathcal{A}$ followed by the median value across units, and $\text{Corr}(\cdot, \cdot)$ denotes

the Pearson correlation of the two quantities. $\widetilde{\mathsf{Corr}}(\cdot, \cdot)$ denotes the Spearman-Brown corrected value of the original quantity (see the "Spearman-Brown Correction" section).

Prior to obtaining the model features of the stimuli for linear regression, we preprocessed each stimulus using the image transforms used on the validation set during model training, resizing the shortest edge of the stimulus in both cases to 64 pixels, preserving the aspect ratio of the input stimulus. Specifically, for models trained using the ImageNet dataset, we first resized the shortest edge of the stimulus to 256 pixels, center-cropped the image to $224 \times 224$ pixels, and finally resized the stimulus to $64 \times 64$ pixels. For models trained using the CIFAR-10 dataset, this resizing yielded a $64 \times 81$ pixels stimulus.

**Representational Similarity Analysis (RSA).** In line with prior work [18, 36], we also used representational similarity analysis (RSA; [30]) to compare models to neural responses, as well as to compare animals to each other. Specifically, we compared (via Pearson correlation) only the upper-right triangles of the representational dissimilarity matrices (RDMs), excluding the diagonals to avoid illusory effects [69].

For each visual area and a given model, we defined the predictivity of the model for that area to be the maximum RSA score across model layers after the suitable noise correction is applied, which is defined as follows. Let $r^\ell$ be the model responses at model layer $\ell$ and let s be the trial-averaged pseudo-population response (i.e., responses aggregated across specimens). The metric used here is a specific instance of Eq (10), where the single source animal A is the trial-wise, deterministic model features (which have a mapping consistency of 1 as a result) and a single target animal B, which is the pseudo-population response:

$$\max_{\ell \in L} \left\langle \frac{\mathsf{RSA}(r^\ell, s_2)}{\sqrt{\widetilde{\mathsf{RSA}}(s_1, s_2)}} \right\rangle,$$

$$\widetilde{\mathsf{RSA}}(s_1, s_2) := \frac{2\, \mathsf{RSA}(s_1, s_2)}{1 + \mathsf{RSA}(s_1, s_2)},$$

$$(1)$$

where $L$ is the set of model layers, $\{s_i\}_{i=1}^2$ are the animal's responses for two halves of the trials (and averaged across the trials dimension), the average is computed over 100 bootstrapped split-half trials, and $\widetilde{\mathsf{RSA}}(s_1, s_2)$ denotes Spearman-Brown correction applied to the internal consistency quantity, $\mathsf{RSA}(s_1, s_2)$, defined in the "Spearman-Brown Correction" section.

If the fits are performed separately for each animal, then B corresponds to each animal among those for a given visual area (defined by the set $\mathcal{A}$), and we compute the median across animals $\mathsf{B} \in \mathcal{A}$:

$$\max_{\ell \in L} \operatorname*{median}_{\mathsf{B} \in \mathcal{A}} \left\langle \frac{\mathsf{RSA}(r^\ell, s_2^\mathsf{B})}{\sqrt{\widetilde{\mathsf{RSA}}(s_1^\mathsf{B}, s_2^\mathsf{B})}} \right\rangle.$$

$$(2)$$

Similar to the above, Spearman-Brown correction is applied to the internal consistency quantity, $\mathsf{RSA}(s_1^\mathsf{B}, s_2^\mathsf{B})$.

## Inter-animal consistency derivation

**Single animal pair.** Suppose we have neural responses from two animals A and B. Let $t_i^p$ be the vector of true responses (either at a given time bin or averaged across a set of time bins) of animal $p \in \mathcal{A} = \{\mathsf{A}, \mathsf{B}, \ldots\}$ on stimulus set $i \in \{\mathsf{train}, \mathsf{test}\}$. Of course, we only receive noisy observations of $t_i^p$, so let $s_{j,i}^p$ be the $j$th set of $n$ trials of $t_i^p$. Finally, let $M(x; y)_i$ be the predictions of a mapping $M$ (e.g., PLS) when trained on input $x$ to match output $y$ and tested on

stimulus set $i$. For example, $M\left(t_{train}^A; t_{train}^B\right)_{test}$ is the prediction of mapping $M$ on the test set stimuli trained to match the true neural responses of animal B given, as input, the true neural responses of animal A on the train set stimuli. Similarly, $M\left(s_{1,train}^A; s_{1,train}^B\right)_{test}$ is the prediction of mapping $M$ on the test set stimuli trained to match the trial-average of noisy sample 1 on the train set stimuli of animal B given, as input, the trial-average of noisy sample 1 on the train set stimuli of animal A.

With these definitions in hand, the inter-animal mapping consistency from animal A to animal B corresponds to the following true quantity to be estimated:

$$\text{Corr}\left(M(t_{train}^A; t_{train}^B)_{test}, t_{test}^B\right), \tag{3}$$

where $\text{Corr}\left(\cdot, \cdot\right)$ is the Pearson correlation across a stimulus set. In what follows, we will argue that Eq (3) can be approximated with the following ratio of measurable quantities, where we split in half and average the noisy trial observations, indexed by 1 and by 2:

$$\text{Corr}\left(M(t_{train}^A; t_{train}^B)_{test}, t_{test}^B\right)$$

$$\sim \frac{\text{Corr}\left(M(s_{1,train}^A; s_{1,train}^B)_{test}, s_{2,test}^B\right)}{\sqrt{\text{Corr}\left(M(s_{1,train}^A; s_{1,train}^B)_{test}, M(s_{2,train}^A; s_{2,train}^B)_{test}\right) \times \text{Corr}\left(s_{1,test}^B, s_{2,test}^B\right)}}. \tag{4}$$

In words, the inter-animal consistency (i.e., the quantity on the left side of Eq (4)) corresponds to the predictivity of the mapping on the test set stimuli from animal A to animal B on two different (averaged) halves of noisy trials (i.e., the numerator on the right side of Eq (4)), corrected by the square root of the mapping reliability on animal A's responses to the test set stimuli on two different halves of noisy trials multiplied by the internal consistency of animal B.

We justify the approximation in Eq (4) by gradually replacing the true quantities (t) by their measurable estimates (s), starting from the original quantity in Eq (3). First, we make the approximation that:

$$\text{Corr}\left(M(t_{train}^A; t_{train}^B)_{test}, s_{2,test}^B\right) \sim \text{Corr}\left(M(t_{train}^A; t_{train}^B)_{test}, t_{test}^B\right) \times \text{Corr}\left(t_{test}^B, s_{2,test}^B\right), \tag{5}$$

by the transitivity of positive correlations (which is a reasonable assumption when the number of stimuli is large). Next, by transitivity and normality assumptions in the structure of the noisy estimates and since the number of trials ($n$) between the two sets is the same, we have that:

$$\begin{aligned} \text{Corr}\left(s_{1,test}^B, s_{2,test}^B\right) &\sim \text{Corr}\left(s_{1,test}^B, t_{test}^B\right) \times \text{Corr}\left(t_{test}^B, s_{2,test}^B\right) \\ &\sim \text{Corr}(t_{test}^B, s_{2,test}^B)^2. \end{aligned} \tag{6}$$

In words, Eq (6) states that the correlation between the average of two sets of noisy observations of $n$ trials each is approximately the square of the correlation between the true value and average of one set of $n$ noisy trials. Therefore, combining Eqs (5) and (6), it follows that:

$$\text{Corr}\left(M(t_{train}^A; t_{train}^B)_{test}, t_{test}^B\right) \sim \frac{\text{Corr}\left(M(t_{train}^A; t_{train}^B)_{test}, s_{2,test}^B\right)}{\sqrt{\text{Corr}\left(s_{1,test}^B, s_{2,test}^B\right)}}. \tag{7}$$

From the right side of Eq (7), we can see that we have removed $t_{test}^B$, but we still need to remove the $M\left(t_{train}^A; t_{train}^B\right)_{test}$ term, as this term still contains unmeasurable (i.e., true) quantities. We apply the same two steps, described above, by analogy, though these approximations may

not always be true (they are, however, true for Gaussian noise):

$$\text{Corr}\left(M(s^A_{1,\text{train}}; s^B_{1,\text{train}})_{\text{test}}, s^B_{2,\text{test}}\right) \sim \text{Corr}\left(s^B_{2,\text{test}}, M(t^A_{\text{train}}; t^B_{\text{train}})_{\text{test}}\right)$$

$$\times \text{Corr}\left(M(t^A_{\text{train}}; t^B_{\text{train}})_{\text{test}}, M(s^A_{1,\text{train}}; s^B_{1,\text{train}})_{\text{test}}\right)$$

$$\text{Corr}\left(M(s^A_{1,\text{train}}; s^B_{1,\text{train}})_{\text{test}}, M(s^A_{2,\text{train}}; s^B_{2,\text{train}})_{\text{test}}\right)$$

$$\sim \text{Corr}\left(M(s^A_{1,\text{train}}; s^B_{1,\text{train}})_{\text{test}}, M(t^A_{\text{train}}; t^B_{\text{train}})_{\text{test}}\right)^2,$$

which taken together implies the following:

$$\text{Corr}\left(M(t^A_{\text{train}}; t^B_{\text{train}})_{\text{test}}, s^B_{2,\text{test}}\right) \sim \frac{\text{Corr}\left(M(s^A_{1,\text{train}}; s^B_{1,\text{train}})_{\text{test}}, s^B_{2,\text{test}}\right)}{\sqrt{\text{Corr}\left(M\left(s^A_{1,\text{train}}; s^B_{1,\text{train}}\right)_{\text{test}}, M\left(s^A_{2,\text{train}}; s^B_{2,\text{train}}\right)_{\text{test}}\right)}}. \tag{8}$$

Eqs (7) and (8) together imply the final estimated quantity given in Eq (4).

**Multiple animals.** For multiple animals, we consider the average of the true quantity for each target in B in Eq (3) across source animals A in the ordered pair (A, B) of animals A and B:

$$\langle\text{Corr}\left(M(t^A_{\text{train}}; t^B_{\text{train}})_{\text{test}}, t^B_{\text{test}}\right)\rangle_{A\in\mathcal{A}:(A,B)\in\mathcal{A}\times\mathcal{A}}$$

$$\sim \left\langle \frac{\text{Corr}\left(M(s^A_{1,\text{train}}; s^B_{1,\text{train}})_{\text{test}}, s^B_{2,\text{test}}\right)}{\sqrt{\widetilde{\text{Corr}}(M(s^A_{1,\text{train}}; s^B_{1,\text{train}})_{\text{test}}, M(s^A_{2,\text{train}}; s^B_{2,\text{train}})_{\text{test}}) \times \widetilde{\text{Corr}}(s^B_{1,\text{test}}, s^B_{2,\text{test}})}} \right\rangle_{A\in\mathcal{A}:(A,B)\in\mathcal{A}\times\mathcal{A}}.$$

We also bootstrap across trials, and have multiple train/test splits, in which case the average on the right hand side of the equation includes averages across these as well.

Note that each neuron in our analysis will have this single average value associated with it when *it* was a target animal (B), averaged over source animals/subsampled source neurons, bootstrapped trials, and train/test splits. This yields a vector of these average values, which we can take median and standard error of the mean (s.e.m.) over, as we do with standard explained variance metrics.

**RSA.** We can extend the above derivations to other commonly used metrics for comparing representations that involve correlation. Since $\text{RSA}(x, y) := \text{Corr}\left(\text{RDM}(x), \text{RDM}(y)\right)$, then the corresponding quantity in Eq (4) analogously (by transitivity of positive correlations) becomes:

$$\langle\text{RSA}(M(t^A_{\text{train}}; t^B_{\text{train}})_{\text{test}}, t^B_{\text{test}})\rangle_{A\in\mathcal{A}:(A,B)\in\mathcal{A}\times\mathcal{A}}$$

$$\sim \left\langle \frac{\text{RSA}(M(s^A_{1,\text{train}}; s^B_{1,\text{train}})_{\text{test}}, s^B_{2,\text{test}})}{\sqrt{\widetilde{\text{RSA}}(M(s^A_{1,\text{train}}; s^B_{1,\text{train}})_{\text{test}}, M(s^A_{2,\text{train}}; s^B_{2,\text{train}})_{\text{test}}) \times \widetilde{\text{RSA}}(s^B_{1,\text{test}}, s^B_{2,\text{test}})}} \right\rangle_{A\in\mathcal{A}:(A,B)\in\mathcal{A}\times\mathcal{A}}. \tag{9}$$

Note that in this case, each *animal* (rather than neuron) in our analysis will have this single average value associated with it when *it* was a target animal (B) (since RSA is computed over images and neurons), where the average is over source animals/subsampled source neurons, bootstrapped trials, and train/test splits. This yields a vector of these average values, which we can take median and s.e.m. over, across animals $B \in \mathcal{A}$.

For RSA, we can use the identity mapping (since RSA is computed over neurons as well, the number of neurons between source and target animal can be different to compare them with

the identity mapping). As parameters are not fit, we can choose train = test, so that Eq (9) becomes:

$$\langle \mathsf{RSA}(t^A, t^B) \rangle_{A \in \mathcal{A}:(A,B) \in \mathcal{A} \times \mathcal{A}} \sim \left\langle \frac{\mathsf{RSA}(s_1^A, s_2^B)}{\sqrt{\widetilde{\mathsf{RSA}}(s_1^A, s_2^A) \times \widetilde{\mathsf{RSA}}(s_1^B, s_2^B)}} \right\rangle_{A \in \mathcal{A}:(A,B) \in \mathcal{A} \times \mathcal{A}}. \tag{10}$$

**Pooled source animal.** Often times, we may not have enough neurons per animal to ensure that the estimated inter-animal consistency in our data closely matches the "true" inter-animal consistency. In order to address this issue, we holdout one animal at a time and compare it to the pseudo-population aggregated across units from the remaining animals, as opposed to computing the consistencies in a pairwise fashion. Thus, B is still the target heldout animal as in the pairwise case, but now the average over A is over a sole "pooled" source animal constructed from the pseudo-population of the remaining animals.

**Spearman-brown correction.** The Spearman-Brown correction can be applied to each of the terms in the denominator individually, as they are each correlations of observations from half the trials of the *same* underlying process to itself (unlike the numerator). Namely,

$$\widetilde{\mathsf{Corr}}(X, Y) := \frac{2\,\mathsf{Corr}\,(X, Y)}{1 + \mathsf{Corr}\,(X, Y)}.$$

Analogously, since $\mathsf{RSA}(X, Y) := \mathsf{Corr}\,(\mathsf{RDM}(x), \mathsf{RDM}(y))$, then we define

$$\begin{aligned}
\widetilde{\mathsf{RSA}}(X, Y) &:= \widetilde{\mathsf{Corr}}(\mathsf{RDM}(x), \mathsf{RDM}(y)) \\
&= \frac{2\,\mathsf{RSA}(X, Y)}{1 + \mathsf{RSA}(X, Y)}.
\end{aligned}$$

## StreamNet architecture variants

We developed shallower, multiple-streamed architectures for mouse visual cortex, shown in Fig 2A. There are three main modules in our architecture: shallow, intermediate, and deep. The shallow and deep modules each consist of one convolutional layer and the intermediate module consists of a block of two convolutional layers. Thus, the longest length of the computational graph, excluding the readout module, is four (i.e., 1 + 2 + 1). Depending on the number of parallel streams in the model, the intermediate module would contain multiple branches (in parallel), each receiving input from the shallow module. The outputs of the intermediate modules are then passed through one convolutional operation (deep module). Finally, the outputs of each parallel branch would be summed together, concatenated across the channels dimension, and used as input for the readout module.

The readout module consists of an (adaptive) average pooling operation that upsamples the inputs to $6 \times 6$ feature maps. These feature maps are then flattened, so that each image only has a single feature vector. These feature vectors are then fed into a linear layer (i.e., fully-connected layer) either for classification or for embedding them into a lower-dimensional space for the contrastive losses.

The table below describes the parameters of three model variants, each containing one ($N = 1$), two ($N = 2$), or six ($N = 6$) parallel branches. Note that one convolutional layer is denoted by a tuple: (number of filters, filter size, stride, padding). The first convolutional layer has max pooling of stride 2, as in AlexNet. A block of convolutional layers is denoted by a list of tuples, where each tuple in the list corresponds to a single convolutional layer. When a list

of tuples is followed by "$\times N$", this means that the convolutional parameters for each of the $N$ parallel branches are the same.

## Neural network training objectives

In this section, we briefly describe the supervised and self-supervised objectives that were used to train our models.

| Module Name | Output Size | Single ($N = 1$) | Dual ($N = 2$) | Six ($N = 6$) |
|---|---|---|---|---|
| Input | $64 \times 64$ | N/A | N/A | N/A |
| Shallow | $7 \times 7$ | $(64, 11, 4, 2)$ | $(64, 11, 4, 2)$ | $(64, 11, 4, 2)$ |
| Intermediate | $3 \times 3$ | $\begin{bmatrix} (192, 5, 1, 2) \\ (384, 3, 1, 1) \end{bmatrix}$ | $\begin{bmatrix} (192, 5, 1, 2) \\ (384, 3, 1, 1) \end{bmatrix} \times 2$ | $\begin{bmatrix} (192, 5, 1, 2) \\ (384, 3, 1, 1) \end{bmatrix} \times 6$ |
| Deep | $3 \times 3$ | If inputs are from intermediate: $(256, 3, 1, 1)$, otherwise: $(256, 3, 2, 0)$ | If inputs are from intermediate: $(256, 3, 1, 1)$, otherwise: $(256, 3, 2, 0)$ | If inputs are from intermediate: $(256, 3, 1, 1)$, otherwise: $(256, 3, 2, 0)$ |

**Supervised training objective.** The loss function $\mathcal{L}$ used in supervised training is the cross-entropy loss, defined as follows:

$$\mathcal{L}(\boldsymbol{X}; \boldsymbol{\theta}) = -\frac{1}{N} \sum_{i=1}^{N} \log \left( \frac{\exp(\boldsymbol{X}_i[\boldsymbol{c}_i])}{\sum_{j=0}^{C-1} \exp(\boldsymbol{X}_i[j])} \right), \tag{11}$$

where $N$ is the batch size, $C$ is the number of categories for the dataset, $\boldsymbol{X} \in \mathbb{R}^{N \times C}$ are the model outputs (i.e., logits) for the $N$ images, $\boldsymbol{X}_i \in \mathbb{R}^C$ are the logits for the $i$th image, $\boldsymbol{c}_i \in [0, C-1]$ is the category index of the $i$th image (zero-indexed), and $\boldsymbol{\theta}$ are the model parameters. Eq (11) was minimized using stochastic gradient descent (SGD) with momentum [70].

**ImageNet** [19]. This dataset contains approximately 1.3 million images in the train set and 50 000 images in the validation set. Each image was previously labeled into $C = 1000$ distinct categories.

**CIFAR-10** [39]. This dataset contains 50 000 images in the train set and 10 000 images in the validation set. Each image was previously labeled into $C = 10$ distinct categories.

**Depth prediction** [40]. The goal of this objective is to predict the depth map of an image. We used a synthetically generated dataset of images known as PBRNet [40]. It contains approximately 500 000 images and their associated depth maps. Similar to the loss function used in the sparse autoencoder objective, we used a mean-squared loss to train the models. The output (i.e., depth map) was generated using a mirrored version of each of our StreamNet variants. In order to generate the depth map, we appended one final convolutional layer onto the output of the mirrored architecture in order to downsample the three image channels to one image channel. During training, random crops of size $224 \times 224$ pixels were applied to the image and depth map (which were both subsequently resized to $64 \times 64$ pixels). In addition, both the image and depth map were flipped horizontally with probability 0.5. Finally, prior to the application of the loss function, each depth map was normalized such that the mean and standard deviation across pixels were zero and one respectively.

Each of our single-, dual-, and six-stream variants were trained using a batch size of 256 for 50 epochs using SGD with momentum of 0.9, and weight decay of 0.0001. The initial learning rate was set to $10^{-4}$ and was decayed by a factor of 10 at epochs 15, 30, and 45.

**Self-supervised training objectives.   Sparse autoencoder** [42]. The goal of this objective is to reconstruct an image from a sparse image embedding. In order to generate an image reconstruction, we used a mirrored version of each of our StreamNet variants. Concretely, the loss function was defined as follows:

$$\mathcal{L}(\boldsymbol{x}; \boldsymbol{\theta}) = \frac{1}{2 \cdot 64^2} \|f(\boldsymbol{x}) - \boldsymbol{x}\|_2^2 + \frac{\lambda}{128} \|\boldsymbol{v}\|_1, \tag{12}$$

where $\boldsymbol{v} \in \mathbb{R}^{128}$ is the image embedding, $f$ is the (mirrored) model, $f(\boldsymbol{x})$ is the image reconstruction, $\boldsymbol{x}$ is a $64 \times 64$ pixels image, $\lambda$ is the regularization coefficient, and $\boldsymbol{\theta}$ are the model parameters.

Our single-, dual-, and six-stream variants were trained using a batch size of 256 for 100 epochs using SGD with momentum of 0.9 and weight decay of 0.0005. The initial learning rate was set to 0.01 for the single- and dual-stream variants and was set to 0.001 for the six-stream variant. The learning rate was decayed by a factor of 10 at epochs 30, 60, and 90. For all the StreamNet variants, the embedding dimension was set to 128 and the regularization coefficient was set to 0.0005.

**RotNet** [44]. The goal of this objective is to predict the rotation of an image. Each image of the ImageNet dataset was rotated four ways (0˚, 90˚, 180˚, 270˚) and the four rotation angles were used as "pseudo-labels" or "categories". The cross-entropy loss was used with these pseudo-labels as the training objective (i.e., Eq (11) with $C = 4$).

Our single-, dual-, and six-stream variants were trained using a batch size of 192 (which is effectively a batch size of $192 \times 4 = 768$ due to the four rotations for each image) for 50 epochs using SGD with Nesterov momentum of 0.9, and weight decay of 0.0005. An initial learning rate of 0.01 was decayed by a factor of 10 at epochs 15, 30, and 45.

**Instance recognition** [45]. The goal of this objective is to be able to differentiate between embeddings of augmentations of one image from embeddings of augmentations of other images. Thus, this objective function is an instance of the class of contrastive objective functions.

A random image augmentation is first performed on each image of the ImageNet dataset (random resized cropping, random grayscale, color jitter, and random horizontal flip). Let $\boldsymbol{x}$ be an image augmentation, and $f(\cdot)$ be the model backbone composed with a one-layer linear multi-layer perceptron (MLP) of size 128. The image is then embedded onto a 128-dimensional unit-sphere as follows:

$$\boldsymbol{z} = f(\boldsymbol{x})/\|f(\boldsymbol{x})\|_2, \qquad \boldsymbol{z} \in \mathbb{R}^{128}.$$

Throughout model training, a memory bank containing embeddings for each image in the train set is maintained (i.e., the size of the memory bank is the same as the size of the train set). The embedding $\boldsymbol{z}$ will be "compared" to a subsample of these embeddings. Concretely, the loss function $\mathcal{L}$ for one image $\boldsymbol{x}$ is defined as follows:

$$
\begin{aligned}
h(\boldsymbol{u}) \quad &= \frac{\exp(\boldsymbol{u} \cdot \boldsymbol{z}/\tau)/Z}{\exp(\boldsymbol{u} \cdot \boldsymbol{z}/\tau)/Z + (m/N)}, \\
\mathcal{L}(\boldsymbol{x}; \boldsymbol{\theta}) \quad &= -\log h(\boldsymbol{v}) - \sum_{j=1}^{m} \log\left(1 - h(\boldsymbol{v}_j)\right),
\end{aligned}
\tag{13}
$$

where $\boldsymbol{v} \in \mathbb{R}^{128}$ is the embedding for image $\boldsymbol{x}$ that is currently stored in the memory bank, $N$ is the size of the memory bank, $m = 4096$ is the number of "negative" samples used, $\{\boldsymbol{v}_j\}_{j=1}^{m}$ are the negative embeddings sampled from the memory bank uniformly, $Z$ is some normalization

constant, $\tau = 0.07$ is a temperature hyperparameter, and $\boldsymbol{\theta}$ are the parameters of $f$. From Eq (13), we see that we want to maximize $h(\boldsymbol{v})$, which corresponds to maximizing the similarity between $\boldsymbol{v}$ and $\boldsymbol{z}$ (recall that $\boldsymbol{z}$ is the embedding for $\boldsymbol{x}$ obtained using $f$). We can also see that we want to maximize $1 - h(\boldsymbol{v}_j)$ (or minimize $h(\boldsymbol{v}_j)$). This would correspond to minimizing the similarity between $\boldsymbol{v}_j$ and $\boldsymbol{z}$ (recall that $\boldsymbol{v}_j$ are the negative embeddings).

After each iteration of training, the embeddings for the current batch are used to update the memory bank (at their corresponding positions in the memory bank) via a momentum update. Concretely, for image $\boldsymbol{x}$, its embedding in the memory bank $\boldsymbol{v}$ is updated using its current embedding $\boldsymbol{z}$ as follows:

$$\boldsymbol{v} \leftarrow \lambda\boldsymbol{v} + (1 - \lambda)\boldsymbol{z},$$
$$\boldsymbol{v} \leftarrow \boldsymbol{v}/\|\boldsymbol{v}\|_2,$$

where $\lambda = 0.5$ is the momentum coefficient. The second operation on $\boldsymbol{v}$ is used to project $\boldsymbol{v}$ back onto the 128-dimensional unit sphere.

Our single-, dual-, and six-stream variants were trained using a batch size of 256 for 200 epochs using SGD with momentum of 0.9, and weight decay of 0.0005. An initial learning rate of 0.03 was decayed by a factor of 10 at epochs 120 and 160.

**MoCov2** [47, 71]. The goal of this objective is to be able to distinguish augmentations of one image (i.e., by labeling them as "positive") from augmentations of other images (i.e., by labeling them as "negative"). Intuitively, embeddings of different augmentations of the same image should be more "similar" to each other than to embeddings of augmentations of other images. Thus, this algorithm is another instance of the class of contrastive objective functions and is similar conceptually to instance recognition.

Two image augmentations are first generated for each image in the ImageNet dataset by applying random resized cropping, color jitter, random grayscale, random Gaussian blur, and random horizontal flips. Let $\boldsymbol{x}_1$ and $\boldsymbol{x}_2$ be the two augmentations for one image. Let $f_q(\cdot)$ be a query encoder, which is a model backbone composed with a two-layer non-linear MLP of dimensions 2048 and 128 respectively and let $f_k(\cdot)$ be a key encoder, which has the same architecture as $f_q$. $\boldsymbol{x}_1$ is encoded by $f_q$ and $\boldsymbol{x}_2$ is encoded by $f_k$ as follows:

$$\boldsymbol{v} = f_q(\boldsymbol{x}_1), \qquad \boldsymbol{k}_0 = f_k(\boldsymbol{x}_2), \qquad \boldsymbol{v}, \boldsymbol{k}_0 \in \mathbb{R}^{128}.$$

During each iteration of training, a dictionary of size $K$ of image embeddings obtained from previous iterations is maintained (i.e., the dimensions of the dictionary are $K \times 128$). The image embeddings in this dictionary are used as "negative" samples. The loss function $\mathcal{L}$ for one image of a batch is defined as follows:

$$\mathcal{L}(\boldsymbol{x}_1, \boldsymbol{x}_2; \boldsymbol{\theta}_q) = -\log \frac{\exp(\boldsymbol{v} \cdot \boldsymbol{k}_0/\tau)}{\sum_{i=0}^{K} \exp(\boldsymbol{v} \cdot \boldsymbol{k}_i/\tau)}, \tag{14}$$

where $\boldsymbol{\theta}_q$ are the parameters of $f_q$, $\tau = 0.2$ is a temperature hyperparameter, $K = 65\,536$ is the number of "negative" samples, and $\{\boldsymbol{k}_i\}_{i=1}^{K}$ are the embeddings of the negative samples (i.e., the augmentations for other images which are encoded using $f_k$, and are stored in the dictionary). From Eq (14), we see that we want to maximize $\boldsymbol{v} \cdot \boldsymbol{k}_0$, which corresponds to maximizing the similarity between the embeddings of the two augmentations of an image.

After each iteration of training, the dictionary of negative samples is enqueued with the embeddings from the most recent iteration, while embeddings that have been in the dictionary for the longest are dequeued. Finally, the parameters $\boldsymbol{\theta}_k$ of $f_k$ are updated via a momentum

update, as follows:

$$\boldsymbol{\theta}_k \leftarrow \lambda\boldsymbol{\theta}_k + (1-\lambda)\boldsymbol{\theta}_q,$$

where $\lambda = 0.999$ is the momentum coefficient. Note that only $\boldsymbol{\theta}_q$ are updated with back-propagation.

Our single-, dual-, and six-stream variants were trained using a batch size of 512 for 200 epochs using SGD with momentum of 0.9, and weight decay of 0.0005. An initial learning rate of 0.06 was used, and the learning rate was decayed to 0.0 using a cosine schedule (with no warm-up).

**SimCLR** [46]. The goal of this objective is conceptually similar to that of MoCov2, where the embeddings of augmentations of one image should be distinguishable from the embeddings of augmentations of other images. Thus, SimCLR is another instance of the class of contrastive objective functions.

Similar to other contrastive objective functions, two image augmentations are first generated for each image in the ImageNet dataset (by using random cropping, random horizontal flips, random color jittering, random grayscaling and random Gaussian blurring). Let $f(\cdot)$ be the model backbone composed with a two-layer non-linear MLP of dimensions 2048 and 128 respectively. The two image augmentations are first embedded into a 128-dimensional space and normalized:

$$\boldsymbol{z}_1 = f(\boldsymbol{x}_1)/\|f(\boldsymbol{x}_1)\|_2, \qquad \boldsymbol{z}_2 = f(\boldsymbol{x}_2)/\|f(\boldsymbol{x}_2)\|_2, \qquad \boldsymbol{z}_1, \boldsymbol{z}_2 \in \mathbb{R}^{128}.$$

The loss function $\mathcal{L}$ for a single pair of augmentations of an image is defined as follows:

$$\mathcal{L}(\boldsymbol{x}_1, \boldsymbol{x}_2; \boldsymbol{\theta}) = -\log\frac{\exp(\boldsymbol{z}_1 \cdot \boldsymbol{z}_2/\tau)}{\sum_{i=1}^{2N} \mathbb{1}[i \neq 1]\exp(\boldsymbol{z}_1 \cdot \boldsymbol{z}_i/\tau)}, \tag{15}$$

where $\tau = 0.1$ is a temperature hyperparameter, $N$ is the batch size, $\mathbb{1}[i \neq 1]$ is equal to 1 if $i \neq 1$ and 0 otherwise, and $\boldsymbol{\theta}$ are the parameters of $f$. The loss defined in Eq (15) is computed for every pair of images in the batch (including their augmentations) and subsequently averaged.

Our single-, dual-, and six-stream variants were trained using a batch size of 4096 for 200 epochs using layer-wise adaptive rate scaling (LARS; [72]) with momentum of 0.9, and weight decay of $10^{-6}$. An initial learning rate of 4.8 was used and decayed to 0.0 using a cosine schedule. A linear warm-up of 10 epochs was used for the learning rate with warm-up ratio of 0.0001.

**SimSiam** [48]. The goal of this objective is to maximize the similarity between the embeddings of two augmentations of the same image. Thus, SimSiam is another instance of the class of contrastive objective functions.

Two random image augmentations (i.e., random resized crop, random horizontal flip, color jitter, random grayscale, and random Gaussian blur) are first generated for each image in the ImageNet dataset. Let $\boldsymbol{x}_1$ and $\boldsymbol{x}_2$ be the two augmentations of the same image, $f(\cdot)$ be the model backbone, $g(\cdot)$ be a three-layer non-linear MLP, and $h(\cdot)$ be a two-layer non-linear MLP. The three-layer MLP has hidden dimensions of 2048, 2048, and 2048. The two-layer MLP has hidden dimensions of 512 and 2048 respectively. Let $\boldsymbol{\theta}$ be the parameters for $f$, $g$, and $h$. The loss function $\mathcal{L}$ for one image $\boldsymbol{x}$ of a batch is defined as follows (recall that $\boldsymbol{x}_1$ and $\boldsymbol{x}_2$ are two augmentations of one image):

$$\boldsymbol{p}_1 = h \circ g \circ f(\boldsymbol{x}_1), \qquad \boldsymbol{p}_2 = h \circ g \circ f(\boldsymbol{x}_2), \qquad \boldsymbol{z}_1 = g \circ f(\boldsymbol{x}_1), \qquad \boldsymbol{z}_2 = g \circ f(\boldsymbol{x}_2),$$

$$\mathcal{L}(\boldsymbol{x}_1, \boldsymbol{x}_2; \boldsymbol{\theta}) = -\frac{1}{2}\left(\frac{\boldsymbol{z}_1 \cdot \boldsymbol{p}_2}{\|\boldsymbol{z}_1\|_2 \|\boldsymbol{p}_2\|_2} + \frac{\boldsymbol{z}_2 \cdot \boldsymbol{p}_1}{\|\boldsymbol{z}_2\|_2 \|\boldsymbol{p}_1\|_2}\right), \tag{16}$$

where $z_1, z_2, p_1, p_2 \in \mathbb{R}^{2048}$. Note that $z_1$ and $z_2$ are treated as constants in this loss function (i.e., the gradients are not back-propagated through $z_1$ and $z_2$). This "stop-gradient" method was key to the success of this objective function.

Our single-, dual-, and six-stream variants were trained using a batch size of 512 for 100 epochs using SGD with momentum of 0.9, and weight decay of 0.0001. An initial learning rate of 0.1 was used, and the learning rate was decayed to 0.0 using a cosine schedule (with no warm-up).

**Barlow twins** [49]. This method is inspired by Horace Barlow's theory that sensory systems reduce redundancy in their inputs [73]. Let $x_1$ and $x_2$ be the two augmentations (random crops and color distortions) of the *same* image, $f(\cdot)$ be the model backbone, and let $h(\cdot)$ be a three-layer non-linear MLP (each of output dimension 8192). Given $z_1, z_2 \in \mathbb{R}^{8192}$, where $z_1 = h \circ f(x_1)$ and $z_2 = h \circ f(x_2)$, this method proposes an objective function which tries to make the cross-correlation matrix computed from the twin embeddings $z_1$ and $z_2$ as close to the identity matrix as possible:

$$z_1 = h \circ f(x_1), \qquad z_2 = h \circ f(x_2),$$
$$\mathcal{L}(x_1, x_2; \theta) = \sum_i (1 - \mathcal{C}_{ii}^2) + \lambda \sum_i \sum_{j \neq i} \mathcal{C}_{ij}^2,$$
$$\mathcal{C}_{ij} := \frac{\sum_b z_{1b,i} z_{2b,j}}{\sqrt{\left(\sum_b z_{1b,i}\right)^2} \sqrt{\left(\sum_b z_{2b,j}\right)^2}}, \tag{17}$$

where $b$ indexes batch examples and $i$, $j$ index the embedding output dimension.

We trained AlexNet (with $64 \times 64$ image inputs) with the recommended hyperparameters of $\lambda = 0.0051$, weight decay of $10^{-6}$, and batch size of 2048 with the LARS [72] optimizer employing learning rate warm-up of 10 epochs under a cosine schedule. We found that training stably completed after 58 epochs for this particular model architecture.

**VICReg** [50]. Let $x_1$ and $x_2$ be the two augmentations (random crops and color distortions) of the *same* image, $f(\cdot)$ be the model backbone, and let $h(\cdot)$ be a three-layer non-linear MLP (each of output dimension 8192). Given $z_1, z_2 \in \mathbb{R}^{8192}$, where $z_1 = h \circ f(x_1)$ and $z_2 = h \circ f(x_2)$, this method proposes an objective function that contains three terms:

1. **Invariance**: minimizes the mean square distance between the embedding vectors.

2. **Variance**: enforces the embedding vectors of samples within a batch to be different via a hinge loss to keep the standard deviation of each embedding variable to be above a given threshold (set to 1).

3. **Covariance**: prevents informational collapse through highly correlated variables by attracting the covariances between every pair of embedding variables towards zero.

We trained AlexNet (with $64 \times 64$ image inputs) with the recommended hyperparameters of weight decay of $10^{-6}$ and batch size of 2048 with the LARS [72] optimizer employing learning rate warm-up of 10 epochs under a cosine schedule, for 1000 training epochs total.

## Top-1 validation set performance

**Performance of primate models on $224 \times 224$ pixels and $64 \times 64$ pixels ImageNet.** Here we report the top-1 validation set accuracy of models trained in a supervised manner on $64 \times 64$ pixels and $224 \times 224$ pixels ImageNet.

| Architecture | Image Size | Objective Function | Top-1 Accuracy |
|---|---|---|---|
| AlexNet | 224 × 224 | Supervised (ImageNet) | 56.52% |
| | 64 × 64 | | 36.22% |
| VGG16 | 224 × 224 | | 71.59% |
| | 64 × 64 | | 58.32% |
| ResNet-18 | 224 × 224 | | 69.76% |
| | 64 × 64 | | 53.31% |
| ResNet-34 | 224 × 224 | | 73.31% |
| | 64 × 64 | | 58.24% |
| ResNet-50 | 224 × 224 | | 76.13% |
| | 64 × 64 | | 63.57% |
| ResNet-101 | 224 × 224 | | 77.37% |
| | 64 × 64 | | 64.53% |
| ResNet-152 | 224 × 224 | | 78.31% |
| | 64 × 64 | | 65.21% |

**Performance of StreamNet variants on 64 × 64 pixels CIFAR-10 and 64 × 64 pixels ImageNet.** Here we report the top-1 validation set accuracy of our model variants trained in a supervised manner on 64 × 64 pixels CIFAR-10 and ImageNet.

| Architecture | Dataset | Objective Function | Top-1 Accuracy |
|---|---|---|---|
| Single Stream | CIFAR-10 | Supervised | 76.52% |
| | ImageNet | | 34.87% |
| Dual Stream | CIFAR-10 | | 81.13% |
| | ImageNet | | 38.68% |
| Six Stream | CIFAR-10 | | 78.73% |
| | ImageNet | | 34.15% |

**Transfer performance of StreamNet variants on 64 × 64 pixels ImageNet under linear evaluation for models trained with self-supervised objectives.** In this subsection, we report the top-1 ImageNet validation set performance under linear evaluation for models trained with self-supervised objectives. After training each model on a self-supervised objective, the model backbone weights are then held fixed and a linear readout head is trained on top of the fixed model backbone. In the case where the objective function is "untrained", model parameters were randomly initialized and held fixed while the linear readout head was trained. The image augmentations used during transfer learning were random cropping and random horizontal flipping. The linear readout for every self-supervised model was trained with the cross-entropy loss function (i.e., Eq (11) with $C = 1000$) for 100 epochs, which was minimized using SGD with momentum of 0.9, and weight decay of $10^{-9}$. The initial learning rate was set to 0.1 and reduced by a factor of 10 at epochs 30, 60, and 90.

## Reinforcement learning task

A set of state-action-reward-state tuples (i.e., $(s_t, a_t, r_t, s_{t+1})$) were generated in prior work [53] and were used to train (in an offline fashion) the reinforcement learning (RL) agent. We used an offline RL algorithm known as critic-regularized regression [56]. Except for the visual

encoder, the architecture of the RL agent was identical to that used by Wang et al. [56] (cf. Fig 3 in Wang et al. [56]). Four different visual encoders based on the AlexNet architecture were used:

- Contrastive ImageNet: AlexNet trained using ImageNet in a contrastive manner (Instance Recognition). Up to the first four layers are implanted into the virtual rodent as its visual system, as these were the layers that best matched mouse visual areas. This visual encoder is the best model of mouse visual cortex (Fig 2). Its weights were held fixed during training of the RL agent.

- Supervised ImageNet: AlexNet trained using ImageNet in a supervised manner. Up to the first four layers are implanted into the virtual rodent as its visual system, as these were the layers that best matched mouse visual areas. Its weights were held fixed during training of the RL agent.

- Contrastive Maze: AlexNet trained using the egocentric maze inputs from the virtual rodent reward-based navigation task. Up to the first four layers are implanted into the virtual rodent as its visual system, as these were the layers that best matched mouse visual areas. Its weights were then held fixed during training of the RL agent.

- Supervised Maze: The first four convolutional layers of AlexNet (ShallowNet) trained end-to-end on the virtual rodent reward-based navigation task.

After training the agent until policy loss convergence by 10 000 steps (once), the agent was evaluated on 300 episodes. Each model was trained twice (i.e., two different random seeds), so each model was evaluated on $300 \times 2 = 600$ episodes in total and we report the average reward across all 600 episodes.

Below, we report the average rewards (and s.e.m. across the 600 episodes) obtained when each of the visual backbones are used by the RL agent to perform the task.

## Evaluating model performance on downstream visual tasks

To evaluate transfer performance on downstream visual tasks, we used the activations from the outputs of the shallow, intermediate, and deep modules of our StreamNet variants. We also included the average-pooling layer in all the variants (the model layer prior to the fully-connected readout layer). The dimensionality of the activations was then reduced to 1000 dimensions using principal components analysis (PCA), if the number of features exceeded 1000. PCA was not used if the number of features was less than or equal to 1000. A linear readout on these features was then used to perform five transfer visual tasks.

For the first four object-centric visual tasks (object categorization, pose estimation, position estimation, and size estimation), we used a stimulus set that was used previously in the evaluation of neural network models of the primate visual system [17, 20, 74]. The stimulus set consists of objects in various poses (object rotations about the $x$, $y$, and $z$ axes), positions (vertical and horizontal coordinates of the object), and sizes, each from eight categories. We then performed five-fold cross-validation on the training split of the medium and high variation image subsets ("Var3" and "Var6", defined by Majaj et al. [75]) consisting of 3840 images, and computed the performance (metrics defined below) on the test split of the medium and high variation sets ("Var3" and "Var6") consisting of 1280 images. Ten different category-balanced train-test splits were randomly selected, and the performance of the best model layer (averaged across train-test splits) was reported for each model. All images were resized to $64 \times 64$ pixels prior to fitting, to account for the visual acuity adjustment. The final non-object-centric task was texture recognition, using the Describable Textures Dataset [58].

**Object categorization.**    We fit a linear support vector classifier to each model layer activations that were transformed via PCA. The regularization parameter,

$$C \in [10^{-8}, 5 \times 10^{-8}, 10^{-7}, 5 \times 10^{-7}, 10^{-6}, 5 \times 10^{-6}, 10^{-5}, 5 \times 10^{-5},$$
$$10^{-4}, 5 \times 10^{-4}, 10^{-3}, 5 \times 10^{-3}, 10^{-2}, 5 \times 10^{-2}, 10^{-1}, 5 \times 10^{-1},$$
$$1, 5, 10^{2}, 5 \times 10^{2}, 10^{3}, 5 \times 10^{3}, 10^{4}, 5 \times 10^{4},$$
$$10^{5}, 5 \times 10^{5}, 10^{6}, 5 \times 10^{6}, 10^{7}, 5 \times 10^{7}, 10^{8}, 5 \times 10^{8}], \tag{18}$$

was chosen by five-fold cross validation. The categories are Animals, Boats, Cars, Chairs, Faces, Fruits, Planes, and Tables. We reported the classification accuracy average across the ten train-test splits.

**Position estimation.**    We predicted both the vertical and the horizontal locations of the object center in the image. We used Ridge regression where the regularization parameter was selected from:

$$\alpha = 1/C, \tag{19}$$

where $C$ was selected from the list defined in (18). For each network, we reported the correlation averaged across both locations for the best model layer.

**Pose estimation.**    This task was similar to the position prediction task except that the prediction target were the $z$-axis (vertical axis) and the $y$-axis (horizontal axis) rotations, both of which ranged between −90 degrees and 90 degrees. The (0, 0, 0) angle was defined in a per-category basis and was chosen to make the (0, 0, 0) angle "semantically" consistent across different categories. We refer the reader to Hong et al. [55] for more details. We used Ridge regression with $\alpha$ chosen from the range in (19).

**Size estimation.**    The prediction target was the three-dimensional object scale, which was used to generate the image in the rendering process. This target varied between 0.625 to 1.6, which was a relative measure to a fixed canonical size of 1. When objects were at the canonical size, they occluded around 40% of the image on the longest axis. We used Ridge regression with $\alpha$ chosen from the range in (19).

**Texture classification.**    We trained linear readouts of the model layers on texture recognition using the Describable Textures Dataset [58], which consists of 5640 images organized according to 47 categories, with 120 images per category. We used ten category-balanced train-test splits, provided by their benchmark. Each split consisted of 3760 train-set images and 1880 test-set images. A linear support vector classifier was then fit with $C$ chosen in the range (18). We reported the classification accuracy average across the ten train-test splits.

## Supporting information

**S1 Fig. StreamNet architecture schematic.** The first four convolutional layers of AlexNet best corresponded to all the mouse visual areas) These convolutional layers were used as the basis for our StreamNet architecture variants. The number of parallel streams, *N*, was varied to be one (single-stream), two (dual-stream) or six (six-stream).
(TIF)

**S2 Fig. Shallow architectures trained with contrastive objective functions yield the best matches to the neural data (calcium imaging dataset).** As in Fig 2, but for the calcium imaging dataset. **A**. The median and s.e.m. neural predictivity, using PLS regression, across neurons in all mouse visual areas except RL. $N = 16228$ units in total (RL is excluded, as mentioned in the "Neural Response Datasets" section). Actual neural predictivity performance can be found in Table 2. The "Primate Model Baseline" denotes a supervised VGG16 trained on 224 px

inputs, used in prior work [14, 15, 36]. All models, except for the "Primate Model Baseline", are trained on 64 px inputs. **B**. Each model's performance on ImageNet is plotted against its median neural predictivity across all units from each visual area. **Inset**. Primate ventral visual stream neural predictivity from BrainScore is correlated with ImageNet categorization accuracy (adapted from Schrimpf et al. [17]). All ImageNet performance numbers can be found in Table 2. Color scheme as in **A** and Fig 2A.
(TIF)

**S3 Fig. Shallow architectures trained with contrastive objective functions yield the best matches to the neural data (RSA). A**. The median and s.e.m. noise-corrected neural predictivity, using RSA, across $N = 39$ and $N = 90$ animals for the Neuropixels and calcium imaging dataset respectively (across all visual areas, with RL excluded for the calcium imaging dataset, as mentioned in the "Neural Response Datasets" section). The "Primate Model Baseline" denotes a supervised VGG16 trained on 224 px inputs, used in prior work [14, 15, 36]). All models, except for the "Primate Model Baseline", are trained on 64 px inputs. **B**. We plot each model's performance on ImageNet against its median neural predictivity, using RSA, across visual areas. All ImageNet performance numbers can be found in Table 2. Color scheme as in **A** and Fig 2A.
(TIF)

**S4 Fig. Structural and functional factors leading to improved neural response predictivity (calcium imaging dataset). A**. As in Fig 3, our dual stream variant (red) and Contrastive AlexNet (brown) were trained using lower resolution ImageNet images and in a contrastive manner. Each image was downsampled from $224 \times 224$ pixels, the image size typically used to train primate ventral stream models, to various image sizes. Training models on resolutions lower than $224 \times 224$ pixels generally led to improved neural predictivity. The median and s.e.m. across neurons in each visual area is reported. As mentioned in the "Neural Response Datasets" section, visual area RL was removed from the calcium imaging neural predictivity results. Refer to Table 1 for $N$ units per visual area. **B**. As in Fig 2B, AlexNet was either untrained, trained in a supervised manner (ImageNet) or trained in an self-supervised manner (instance recognition). We observe that the first four convolutional layers provide the best fits to the neural responses for all the visual areas while the latter three layers are not very predictive for any visual area. As mentioned in the "Neural Response Datasets" section, visual area RL was removed from the calcium imaging neural predictivity results.
(TIF)

**S5 Fig. Inter-animal consistency as a function of the number of units in the datasets for different mapping transforms.** Inter-animal consistency was extrapolated across the number of units in each dataset using a log-linear function ($f(n) = a \log_{10}(n) + b$, where $a$ and $b$ are fit as parameters via least squares, and $n$ is the sample size factor). This analysis reveals that inter-animal consistency of the Neuropixels dataset approaches 1.0 more rapidly than it does for the calcium imaging dataset. Inter-animal consistency evaluated at a sample size factor of one indicates the consistency when all the existing units in the datasets are used (i.e., inter-animal consistency values reported in Fig 1A).
(TIF)

**S6 Fig. Inter-animal consistency can increase with more stimuli. A**. Inter-animal consistency under PLS regression evaluated on the train set (left bars for each visual area) and test set (right bars for each visual area), for both Neuropixels and calcium imaging datasets. The horizontal lines are the internal consistency (split-half reliability). **B**. Inter-animal consistency

under PLS regression on the train set (dotted lines) and test set (straight lines), aggregated across visual areas. Each dot corresponds to the inter-animal consistency evaluated across 10 train-test splits, where each split is a sample of the natural scene image set corresponding to the percentage (x-axis). Note that RL is excluded for calcium imaging, as explained in the text (the "Neural Response Datasets" section). The median and s.e.m. across neurons is reported for both panels. Refer to Table 1 for $N$ units per visual area.
(TIF)

**S7 Fig. Data augmentations alone do not lead to improved neural predictivity.** Here we compare the neural predictivity of AlexNet trained in three different ways. Contrastive ImageNet is an AlexNet trained using instance recognition on ImageNet with augmentations that are part of the contrastive algorithm (random crop, random color jitter, random grayscale, random horizontal flip). Supervised ImageNet is an AlexNet trained on ImageNet in a supervised manner with a smaller set of augmentations (random crop and random horizontal flip). Supervised ImageNet (contrastive augmentations) is an AlexNet trained on ImageNet in a supervised manner with the augmentations used in the instance recognition algorithm. This control model allows us to ascertain whether the improved neural predictivity of the Contrastive ImageNet model (red) is due to the contrastive loss function itself or due to the larger set of image augmentations used during model training. In both neural response datasets, we can conclude that data augmentations alone do not contribute to improved correspondence with the mouse visual areas.
(TIF)

**S8 Fig. Out-of-distribution task performance across model layers.** For the models trained on images from the maze environment, we plot their transfer performance on a set of out-of-distribution tasks (described in lower right panel of Fig 6C) across model layers. We find that intermediate model areas are better able to perform the transfer tasks and that the model layers that attain peak performance on the tasks correspond to those that best predict neural responses in the intermediate/higher mouse visual areas (see Fig 2B).
(TIF)

**S9 Fig. Neural predictivity on the Neuropixels dataset while fixing architecture (left) and objective function (right).** The data in the left panel shows how neural predictivity varies when the architecture is fixed to be AlexNet, but the objective function is varied between supervised (object categorization) and self-supervised, contrastive objectives. The data in the right panel shows how neural predictivity varies when the objective function is fixed to be instance recognition, but the architecture is varied, including StreamNets, MouseNets, VGG16, ResNets, and AlexNet.
(TIF)

**S10 Fig. Brain hierarchy score.** The brain hierarchy score metric of Nonaka et al. [51] was computed for a set of feedforward CNNs. Self-supervised, contrastive models (and shallower models) have a higher brain hierarchy score, computed using the mapping from model features to electrophysiological responses.
(TIF)

## Acknowledgments

We thank Shahab Bakhtiari, Katherine L. Hermann, and Akshay Jagadeesh for helpful discussions, and Eshed Margalit and Xiaoxuan Jia for helpful feedback on an initial draft of the manuscript.

## Author Contributions

**Conceptualization:** Aran Nayebi, Nathan C. L. Kong.

**Data curation:** Aran Nayebi, Nathan C. L. Kong.

**Formal analysis:** Aran Nayebi, Nathan C. L. Kong.

**Funding acquisition:** Anthony M. Norcia, Daniel L. K. Yamins.

**Investigation:** Aran Nayebi, Nathan C. L. Kong.

**Methodology:** Aran Nayebi, Nathan C. L. Kong, Chengxu Zhuang, Justin L. Gardner, Daniel L. K. Yamins.

**Project administration:** Aran Nayebi, Nathan C. L. Kong.

**Resources:** Anthony M. Norcia, Daniel L. K. Yamins.

**Software:** Aran Nayebi, Nathan C. L. Kong.

**Supervision:** Anthony M. Norcia, Daniel L. K. Yamins.

**Validation:** Aran Nayebi, Nathan C. L. Kong.

**Visualization:** Aran Nayebi, Nathan C. L. Kong.

**Writing – original draft:** Aran Nayebi, Nathan C. L. Kong.

**Writing – review & editing:** Aran Nayebi, Nathan C. L. Kong, Chengxu Zhuang, Justin L. Gardner, Anthony M. Norcia, Daniel L. K. Yamins.

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
