## [Decision Letter · Decision Letter 0]

16 May 2023

Dear Dr. Nayebi,

Thank you very much for submitting your manuscript "Mouse visual cortex as a limited resource system that self-learns an ecologically-general representation" for consideration at PLOS Computational Biology.

As with all papers reviewed by the journal, your manuscript was reviewed by members of the editorial board and by several independent reviewers. In light of the reviews (below this email), we would like to invite the resubmission of a significantly-revised version that takes into account the reviewers' comments.

The three reviewers have done a great job at providing in-depth constructive feedback (while agreeing on many points raised). The addressing of their comments will certainly improve this submission and we look forward to reading the resubmission. Please note that "All code can be accessed by emailing the corresponding authors." is not an acceptable form of code/data sharing for this journal (see PLOS Data policy). Code and data need to be shared via public repositories.

We cannot make any decision about publication until we have seen the revised manuscript and your response to the reviewers' comments. Your revised manuscript is also likely to be sent to reviewers for further evaluation.

Sincerely,

Tim Christian Kietzmann, Dr. rer. nat.

Academic Editor

PLOS Computational Biology

Daniele Marinazzo

Section Editor

PLOS Computational Biology

The three reviewers have done a great job at providing in-depth constructive feedback (while agreeing on many points raised). The addressing of their comments will certainly improve this submission and I look forward to reading the resubmission. Please note that "All code can be accessed by emailing the corresponding authors." is not an acceptable form of code/data sharing for this journal (see PLOS Data policy). Code and data need to be shared via public repositories.

Reviewer's Responses to Questions

**Comments to the Authors:**

Reviewer #1: Nayabi, Kong et al. compared the performance of models trained with supervised and self-supervised learning in predicting the responses of mouse visual areas to natural stimuli. Their results showed an outperformance of Shallow architectures trained with SSL compared to other models. In addition, the SSL-trained models showed better out-of-distribution generalization, based on which the authors suggest that the mouse visual cortex can be understood as a shallow general purpose visual system. I enjoyed reading this paper, and believe it offers a strong contribution to the comparison of visual systems across different species using deep neural networks. I have a few concerns that I elaborate below.

Major comments:

1) If the focus of the paper is on evaluating the importance of contrastive methods for predicting the responses of mouse visual areas, it would be important to include comparisons with non-contrastive SSL techniques, such as Barlow Twins, VICReg, and others. Alternatively, if such comparisons are not included, it may be more appropriate to use a different terminology rather than 'contrastive' for SSL models.

2) Regarding Section 2.1, I found it unclear why the mapping method that resulted in the highest inter-animal similarity (PLS) was chosen as the similarity measure, especially considering that the correlation values across different metrics are not that different. It would be helpful to understand the range of these differences and why a higher value is considered indicative of true similarity. Additionally, it would be interesting to know how the results might differ if a different metric, such as CKA, were used instead. It is important to explicitly explain the differences in results obtained with different similarity measures in the main text.

3) The authors employed the Contrastive ShallowNet architecture, which comprises the first four layers of AlexNet trained with contrastive SSL. I wonder why the authors did not solely use a 4-layer convolutional ANN and train it with contrastive SSL if a shallow architecture is sufficient for the purpose of explaining the responses of mouse visual areas.

4) I find the argument presented in Figure 3 for the role of depth in predicting mouse visual cortex responses unconvincing, as there are several confounding factors that make the interpretation difficult. Eg, some models listed in fig 3 are based on AlexNet architecture, while others use ResNet architecture (also some architectures other than AlexNet are listed in fig 1 despite the claim that only AlexNet was used in the experiments). Additionally, some models were trained with contrastive objectives, while others were trained with supervised objectives. Even within the contrastive methods, there are different contrastive objectives. It would be more compelling if the effect of architecture depth was demonstrated by controlling for other factors, such as the objective function.

5) The reward-based navigation task presented in fig 6A requires further explanation. What is the goal of the task? What kind of visual features are informative in solving the task effectively? It is hard to conceive how a model trained on static images would be useful for a dynamic behavior like navigation. Moreover, to better understand the differences between SSL and supervised models in relation to this task, it would be helpful to compare them with a model directly trained with RL on the task in terms of the mean episode return.

6) I found the purpose of the experiments with the virtual rat unclear. If similar experiments were conducted using a virtual monkey simulation, would we expect to observe similar results? How would the interpretations differ? Is there anything specific in the rat's movement repertoire or visual experience that necessitates the use of representations learned with SSL?

7) I noticed that the results of the Neuropixel and calcium imaging data are not entirely consistent. Specifically, the differences between the performance of the best SSL and supervised models versus autoencoding models do not appear to be significant with calcium imaging based on fig S3. Could the authors provide any explanation for these discrepancies? I am curious whether the deconvolved spike timings or calcium traces were used in these experiments.

8) Could the intrinsic dimensionality of the learned representations account for the transferability of the SSL model?

Minor comments:

1) What are error bars when showing models’ predictivity difference e.g. fig 1A)?I suggest that the authors provide more clarity in reporting the statistical differences between the various models.

2) The Color scheme needs to be added to fig 3.

3) The details of the mouse visual cortex data, stimuli etc needs to be included in the results.

Reviewer #2: This work identifies several important features that make a vision model a better fit for mouse versus primate vision. These features reflect known differences between the two systems such as architecture depth and image resolution. My concerns are mainly around clarity and presentation, with some requests for extra analyses.

Clarity:

Given that aspects of interanimal consistency as a metric are discussed throughout, I think it would make sense to include fig S1 or some variant of it in the main text in order to clarify the metrics and to motivate the focus on neuropixels over calcium imaging.

I found the choice to refer to supervised VGG-16 as the "prior primate model" confusing, as many different models have been used to predict neural activity in the primate and giving it this title suggested to me it was somehow more fit to primate data than a standard VGG16. I'm fine with the authors explaining this choice as the baseline due to its common use as a primate model, but I think in the figures and discussion of results it should just be labeled as supervised VGG-16.

In section 2.2, there was not an explicit enough distinction between image set and objective. For example, ImageNet could be considered "human-centric" by the nature of the images themselves, not just the fact that category labels are applied to them, and this aspect of the data would impact both supervised and self-supervised methods. The idea of using fewer categories is also conflated with the type of images (CIFAR-10 is not just fewer categories, it is smaller and lower quality images as well). I would like the description of the models in this section to acknowledge when different image sets are used in addition to different training objectives.

I did not find Figure 3 very valuable. The depth of the overall network is not the same as the number of LNL operations that occur before the units responsible for the neural prediction (as 1B shows much of the predictive power is coming from early layers). It seems the latter should be used if the goal is to show that not many LNLs are needed to predict mouse visual responses.

Relatedly, I am confused as to why the authors choose to refer to their best-performing model as "Contrastive ShallowNet" versus simply Contrastive AlexNet. The architecture of the trained model is AlexNet. The fact that the first four layers are most predictive of mouse response is useful to know, but it does not make the model as it was built any shallower. I imagine many of the other networks also rely primarily on earlier layers for their predictions, but they are not referred to as a separate model as result of this. If the authors showed that a model trained with a shallower architecture was better than I would understand, but as it stands now this name change is confusing and is not applied evenly across all models.

In figure 6, why are not all models evaluated on all tasks? At the very least, it is important to include the reward performance of the 'supervised' maze model in Fig 6A in order to contextualize these numbers. Relatedly, I'd like to see the authors discuss how these results contrast with Lindsay et al, which shows somewhat better match to mouse data with the an RL-trained model over supervised/unsupervised methods (knowing how well the RL trained visual system here works may be important for this comparison)

The authors state "as we find that higher model areas best support these scene understanding transfer tasks". Was this shown somewhere?

Fig 4 inset. Are those all supervised? (they are all black)

"We believe that future work could investigate other appropriate low-pass filters and ecologically-relevant pixel-level transformations to apply to the original image or video stream" - The authors may want to cite some existing work in this direction such as https://www.mdpi.com/2079-9292/10/22/2883 and https://www.biorxiv.org/content/10.1101/2020.06.16.154542v2

I assume the authors will be publishing their code upon publication of the paper (as per PlosCB policy) but right now it says it is only available via email.

Additional analyses:

Looking at the hierarchy score (e.g. https://www.sciencedirect.com/science/article/pii/S2589004221009810), based on the rough hierarchical categorization the authors find in S1B would be a helpful additional analysis that may reveal other trends in the impact of architecture and training. From what has already been plotted it seems possible that self-supervised learning better recapitulates the visual hierarchy as well.

Why is there no self-supervised MouseNet? (also the plotting of MouseNet performance in Figure 1 has some other bar under it)

Reviewer #3: # Summary Of Contributions

The paper presents a computational model of the mouse visual cortex and finds that a shallow network with a low-resolution input is optimal for modeling mouse visual cortex. The authors find that models trained with self-supervised contrastive objectives are better matches to mouse cortex than models trained on supervised objectives or non-contrastive self-supervised methods. They show that the self-supervised, contrastive objective builds a general-purpose visual representation that transfers better to out-of-distribution visual scene understanding and reward-based navigation tasks.

# Strengths And Weaknesses

## Strengths

+ Paper is overall well organized

+ Convincing evidence presented for the main claims

+ Provides a nice comparative view on mouse vs. monkey visual system

## Weaknesses

- There may be an issue with image size and resolution

- Writing and presentation of results can be improved to make the paper more accessible

# Detailed Comments

The paper presents a thoroughly executed study whose main claims are well supported by the evidence presented. I am generally very supportive of the paper and have only two main concerns which I consider crucial to be addressed in a revision:

1. You make a claim about 64 px being the optimal image size for training. This sounds reasonable given the visual acuity and field of view of the mouse you cite in section 2.3. However, the argument cites a visual field of 60–90°, but the images spanned 120° in the experiments (according to the technical white paper on the dataset). Thus, the image is highly distorted, the resolution depends a lot on the eccentricity and I am not quite sure if the math actually works out. It would be great to look at this a bit more carefully.

2. While the paper is overall very well organized, I found the results more difficult to read and digest than necessary and the methods sometimes incomplete. The following examples illustrate my issues, and I would highly encourage the authors to revise sections 2, 3 and 5 from the perspective of a reader who is not already familiar with the work and all the details.

- Section 2.1 is quite confusing. It's not clear to me what this section wants to accomplish. The claim "We identified the best-performing mapping function by assessing a variety of functions to map responses from one animal to those of another" is unclear to me. Why do you find the best-performing mapping function that way? Why would one want to map responses from one animal to another? If this is so central that you start discussing it as the first point after the intro, why don't you show any data supporting your claim (only something hidden in Fig. S1, but I couldn't figure out what exactly it shows and how it supports the claim).

- The text sometimes refers to the "prior primate model" (e.g. Section 2.2, first paragraph), but I don't think this concept has been clearly introduced. I can only guess "prior primate model" means VGG-16?

- The results of Fig. 1A could be presented in a more systematic way. Rather than just sorting by performance and some rough grouping, I think it would be better to systematically investigate individual factors while keeping all others constant. For instance, the best-performing is ShallowNet trained with IR. What happens you replace IR by other objectives? I found a few other contrastive objectives, but where is ImageNet-supervised ShallowNet? Is that the same as supervised AlexNet? If so, why do you call them differently? Similarly, keeping IR as the objective, how does architecture affect performance? I think in this case the relevant data points are shown in Fig. 1A, but it's quite tedious to find them.

- The discussion of objectives in 2.2 is nice, but again I found it difficult to map to Fig. 1A. Categorization with 10 classes (Krizhevsky 2009) probably refers to CIFAR-10, depth prediction (Zhang 2017) probably to the orange bars, sparse autoencoding (Olshausen and Field 1996) probably pink. Why don't you state these things? What are the single-, two- and six-stream architectures and how do they relate to Olshausen and Field 1996 (which doesn't actually use a CNN or even an explicit encoder)?

- It's not quite clear to me whether "Contrastive ShallowNet" refers to just taking the first four layers of an entire AlexNet trained using IR or whether you take only the first four layers of AlexNet and add a global average pooling + fully connected layer in order to train IR. Please make the description a bit more explicit in Section 2.2 and the caption of Fig. 1.

- Related to previous point: I couldn't find a description in the methods how the StreamNet architecture (and ShallowNet) get from (3x3) feature maps to a single latent vector that is needed for, e.g., the contrastive losses or the classification objective. Global average pooling over space? Followed by a fully-connected layer? The sentence "Finally, the outputs of each parallel branch would be summed together, concatenated across the channels dimension, and used as input for the readout module" is very ambiguous? Summed across which dimension? What's the "readout module"?

- The StreamNet architecture is first mentioned in section 2.3, and was not at all clear to me from this section. I had to search for it in the methods to understand the architecture (somewhat). I'm still confused about what you mean by "StreamNet incorporates dense skip connections" - where are these skip connections? They are not mentioned in the methods section 5.4.

- It didn't become clear to me why you can train AlexNet only on 64 px upwards by Contrastive StreamNet can be trained on 32 px. You could simply remove, e.g., the last max pooling layer.

## Other Minor Issues:

- Abstract: second sentence appears broken: "However, an overall understanding of the mouse’s visual cortex, and how it supports a range [OF?] behaviors, remains unknown."

- The claim "We also attain neural predicitivity improvements over prior work...": Where is the data supporting this claim? The sentence references five papers, but I can only find MouseNet in Fig. 1. If some of the models in Fig. 1 map onto the references in this sentence, please state it explicitly somewhere so your readers can make the link.

- Fig. 1 caption: "All models are trained on 64 px inputs unless otherwise stated in figure." It's no where stated otherwise, only explicitly 64 px for VGG-16. Does this mean all models trained on 64 or did you forget to state it somewhere? CIFAR-10 has 32x32 images. Does that mean you're upsampling them to 64?

- You repeatedly make the point that the categories of ImageNet are human-centric and not relevant for mice. But if that's the case, why would ImageNet be a good approach for monkeys, for whom ImageNet categories should be equally irrelevant?

**Have the authors made all data and (if applicable) computational code underlying the findings in their manuscript fully available?**

Reviewer #1: **No: **The models and the analysis codes are not released as supporting information or deposited to a public repository. It is only mentioned that "All code can be accessed by emailing the corresponding authors."

Reviewer #2: **No: **The code is listed as available by email

Reviewer #3: **No: **Statement on code availability reads "All code can be accessed by emailing the corresponding authors." I do not consider that publicly available, as it is not deposited in a publicly accessible repository.

PLOS authors have the option to publish the peer review history of their article (what does this mean?). If published, this will include your full peer review and any attached files.

Reviewer #1: No

Reviewer #2: No

Reviewer #3: No
---

## [Decision Letter · Decision Letter 1]

11 Sep 2023

Dear Dr. Nayebi,

We are pleased to inform you that your manuscript 'Mouse visual cortex as a limited resource system that self-learns an ecologically-general representation' has been provisionally accepted for publication in PLOS Computational Biology.

Before your manuscript can be formally accepted you will need to complete some formatting changes, which you will receive in a follow up email. A member of our team will be in touch with a set of requests. Moreover, we would like to ask you to include the hierarchy score results, as requested by reviewer 2, in the supplement of the paper.

Best regards,

Tim Christian Kietzmann, Dr. rer. nat.

Academic Editor

PLOS Computational Biology

Daniele Marinazzo

Section Editor

PLOS Computational Biology

Reviewer's Responses to Questions

**Comments to the Authors:**

Reviewer #1: I thank the authors for their thorough responses to my comments! All my concerns are addressed. Based on the revisions made, I recommend this manuscript for publication.

Reviewer #2: I am mostly content with the changes, but I would strongly suggest the authors include the hierarchy score results at least in the supplemental material. Capturing the hierarchical relationship is important for use of these networks as *mechanistic models* of visual cortex rather than just predictors of neural activity and it is therefore good to know which models perform well on this metric.

Reviewer #3: I would like to thank the authors for the thorough revision. They have addressed my comments.

**Have the authors made all data and (if applicable) computational code underlying the findings in their manuscript fully available?**

Reviewer #1: Yes

Reviewer #2: None

Reviewer #3: Yes

PLOS authors have the option to publish the peer review history of their article (what does this mean?). If published, this will include your full peer review and any attached files.

Reviewer #1: No

Reviewer #2: No

Reviewer #3: No

---

## [Editor Report · Acceptance letter]

25 Sep 2023

PCOMPBIOL-D-23-00393R1 

Mouse visual cortex as a limited resource system that self-learns an ecologically-general representation

Dear Dr Nayebi,

I am pleased to inform you that your manuscript has been formally accepted for publication in PLOS Computational Biology. Your manuscript is now with our production department and you will be notified of the publication date in due course.

With kind regards,

Timea Kemeri-Szekernyes
